# FLOWALIGN: TRAJECTORY-REGULARIZED, INVERSION-FREE FLOW-BASED IMAGE EDITING

**Jeongsol Kim\*, Yeobin Hong\*, Jonghyun Park, Jong Chul Ye**
KAIST
`{jeongsol, yeobin34, jhpark99, jong.ye}@kaist.ac.kr`
\* Equal contribution

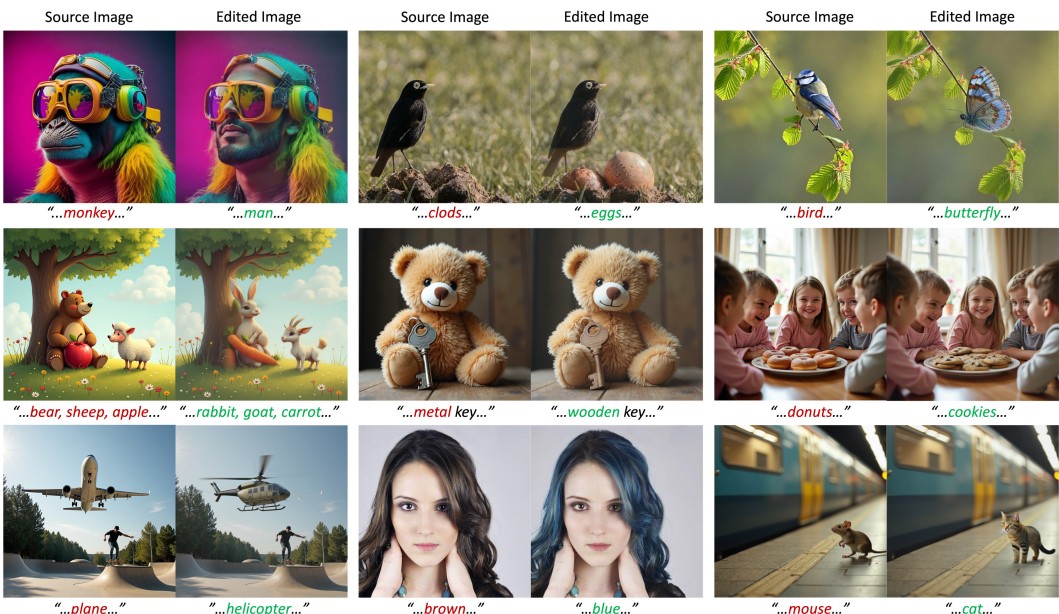

Figure 1: Representative editing results produced by FlowAlign, where the red portion of the prompt is replaced with the green portion. Samples are drawn from EditBench and PIEBench.

## ABSTRACT

Recent inversion-free, flow-based image editing methods such as FlowEdit leverages a pre-trained noise-to-image flow model such as Stable Diffusion 3, enabling text-driven manipulation by solving an ordinary differential equation (ODE). While the lack of exact latent inversion is a core advantage of these methods, it often results in unstable editing trajectories and poor source consistency. To address this limitation, we propose *FlowAlign*, a novel inversion-free flow-based framework for consistent image editing with optimal control-based trajectory control. Specifically, FlowAlign introduces source similarity at the terminal point as a regularization term to promote smoother and more consistent trajectories during the editing process. Notably, our terminal point regularization is shown to explicitly balance semantic alignment with the edit prompt and structural consistency with the source image along the trajectory. Furthermore, FlowAlign naturally supports reverse editing by simply reversing the ODE trajectory, highliting the reversible and consistent nature of the transformation. Extensive experiments demonstrate that FlowAlign outperforms existing methods in both source preservation and editing controllability.

# 1    INTRODUCTION

In text-based image editing (Meng et al., 2021; Mokady et al., 2023; Tumanyan et al., 2023; Hertz et al., 2023; Kim et al., 2024; Kulikov et al., 2024; Brooks et al., 2023), the goal is to transform a source image into a target based on either textual descriptions of the images or specific editing instructions. From a distributional perspective, the image editing task can be interpreted as a continuous normalizing flow (CNF) (Papamakarios et al., 2021) that pushes forward a source distribution to a target distribution. Specifically, we parameterize a velocity field, that uniquely determines the flow, using a neural network. Then, the generative process corresponds to solving an ordinary differential equation (ODE) governed by the trained velocity field. To reduce the computational cost of simulating ODEs during training for likelihood evaluation, flow matching has been proposed (Lipman et al., 2023). Its conditional variant enables direct supervision by computing the target velocity, allowing efficient training of flow models between arbitrary distributions. These flow models includes score-based diffusion models (Song et al., 2021b; Ho et al., 2020; Song et al., 2021a) as well as rectified flow models (Liu et al., 2023b; Esser et al., 2024).

Recently, within the framework of flow-based models, text-to-image generation has achieved significant advancements through improved time discretization, loss weighting, and model architecture, notably based on DiT (Peebles & Xie, 2023). While these foundational models are trained to map samples from a normal distribution to a clean data distribution, additional training—either from scratch or via fine-tuning—is necessary to establish a flow between two arbitrary distributions.

To mitigate this additional cost for constructing a new flow between image distributions, several approaches such as SDEdit (Meng et al., 2021) and Dual Diffusion Implicit Bridge (DDIB) (Su et al., 2023) have leveraged pre-trained noise-to-image diffusion models. However, SDEdit requires careful selection of an appropriate initial noise level for editing, while DDIB relies on an inversion process that is prone to errors arising from discretization and an approximated velocity field for subsequent timesteps.

Recently, RF-inversion (Rout et al., 2025a) proposes an optimal-control based inversion for flow models, in which a guiding vector field steers the dynamics toward high-likelihood samples via velocity interpolation. Unfortunately, it still involves computational overhead due to the ODE inversion. To address this, FlowEdit (Kulikov et al., 2024) proposed simulating an ODE between two image samples without inversion. However, empirical results showed that the method is quite sensitive to the hyperparameters and often fails to retain the source consistency. Moreover, the method heuristically applies classifier-free-guidance (CFG) to the both source and target velocity fields with different scaling factor and relies on skipping early timesteps of the ODE to enhance editing quality. These design choices introduce multiple hyperparameters to be searched and compromise the deterministic nature of ODE (see Section 4).

Motivated by the observation that these limitations of FlowEdit arise from nonsmooth and unstable editing trajectories—partly due to the lack of explicit latent inversion—we introduce *FlowAlign*, an optimal control-based inversion-free approach for consistent and controllable text-driven image editing through trajectory regularization. In contrast to RF-inversion that requires an ODE inversion, we introduce a structural similarity at the terminal point as our trajectory regularization term to overcome the instability caused by the absence of the inverted latent. Although the regularization is primarily enforced at the terminal point, we further observe that it also naturally enforces source consistency along the trajectory by penalizing unnecessary deviations from the original image.

In contrast to methods like FlowEdit, which require four times the NFE due to the reliance on classifier-free guidance on both source and target sides, our approach is more computationally efficient. FlowAlign introduces a minimal overhead of only one additional function evaluation (NFE) per ODE step to apply its regularization term. Despite this efficiency, FlowAlign achieves superior source preservation and competitive or improved editing quality compared to FlowEdit and also other existing approaches in image, video, and 3D editing. Finally, FlowAlign supports editing through backward ODE with the learned flow field, enabling accurate reconstruction of the original image from the edited output. This highlights the reversible and deterministic nature of the learned transformation, made possible by our explicit trajectory regularization.

## 2 BACKGROUNDS

Suppose we have access to samples from two distributions $X_1 \sim p$ and $X_0 \sim q$ that forms independent coupling, $\pi_{0,1}(X_0, X_1) = p(X_1)q(X_0)$. We can define a time-dependent function called flow $\psi_t(\boldsymbol{x}) : [0,1] \times \mathbb{R}^d \to \mathbb{R}^d$, which is diffeomorphism and satisfies $\psi_t(X_1) = X_t$, to describe a continuous transform between $X_1$ and $X_0$. Specifically, for $0 \leq s < t \leq 1$, we can sequentially transfer the sample $X_t$ to $X_s = \psi_s(X_1) = \psi_s(\psi_t^{-1}(X_t)) = \psi_{s|t}(X_t)$. Here, the $\psi_t$ is uniquely characterized by a flow ODE

$$d\psi_t(\boldsymbol{x}) = \boldsymbol{v}_t(\psi_t(\boldsymbol{x}))dt \tag{1}$$

where $\boldsymbol{v}_t$ represents velocity field. Here, $\boldsymbol{v}_t$ is approximated by a neural network $\boldsymbol{v}_t^\theta$ with the parameter $\theta$ to construct a flow as generative model. The training objective for the parameterized velocity field is called flow matching, which is expressed as

$$\mathcal{L}_{FM} = \mathbb{E}_{t \in [0,1], \boldsymbol{x}_t \sim p_t} \|\boldsymbol{v}_t(\boldsymbol{x}_t) - \boldsymbol{v}_t^\theta(\boldsymbol{x}_t)\|^2. \tag{2}$$

Unfortunately, we cannot access $\boldsymbol{v}_t(\boldsymbol{x}_t)$ due to intractable integration over all $\boldsymbol{x}_0$. To address this, Lipman et al. (2023) proposes conditional flow matching:

$$\mathcal{L}_{CFM} = \mathbb{E}_{t \in [0,1], \boldsymbol{x}_0 \sim q} \|\boldsymbol{v}_t(\boldsymbol{x}_t|\boldsymbol{x}_0) - \boldsymbol{v}_t^\theta(\boldsymbol{x}_t)\|^2 \tag{3}$$

where $\boldsymbol{v}_t(\boldsymbol{x}_t|\boldsymbol{x}_0)$ is a conditional velocity field given by

$$\boldsymbol{v}_t(\boldsymbol{x}_t|\boldsymbol{x}_0) = \dot{\psi}_t(\psi_t^{-1}(\boldsymbol{x}_t|\boldsymbol{x}_0)|\boldsymbol{x}_0) = \dot{\psi}_t(\boldsymbol{x}_1|\boldsymbol{x}_0), \tag{4}$$

where the last equality is due to the definition of $\boldsymbol{x}_t = \psi_t(\boldsymbol{x}_1|\boldsymbol{x}_0)$. One of the most important contribution of Lipman et al. (2023) is that the trained velocity field $\boldsymbol{v}_t^\theta(\boldsymbol{x}_t)$ from the conditional flow matching Eq. (3) can also approximate the *unconditional* velocity field $\boldsymbol{v}_t(\boldsymbol{x}_t)$ in Eq. (2). Accordingly, after training of the velocity field using Eq. (3), we can generate a sample by solving the following ODE:

$$d\boldsymbol{x} = \boldsymbol{v}_t^\theta(\boldsymbol{x}_t)dt. \tag{5}$$

Among various flows, the affine conditional flow defined as $\psi_t(\boldsymbol{x}_1|\boldsymbol{x}_0) = a_t\boldsymbol{x}_0 + b_t\boldsymbol{x}_1$ is widely used, where $a_0 = b_1 = 1$ and $a_1 = b_0 = 0$. The resulting conditional velocity field derived from the rectified flow for Eq. (3) is then given by

$$\boldsymbol{v}_t(\boldsymbol{x}_t|\boldsymbol{x}_0) = \dot{\psi}_t(\psi_t^{-1}(\boldsymbol{x}_t|\boldsymbol{x}_0)|\boldsymbol{x}_0) = \dot{a}_t\boldsymbol{x}_0 + \dot{b}_t\boldsymbol{x}_1. \tag{6}$$

In case of linear conditional flow (or rectified flow) (Liu et al., 2023b), $a_t = t$ and $b_t = 1 - t$, the conditional velocity field derived from the rectified flow for Eq. (3) is $\boldsymbol{v}_t(\boldsymbol{x}_t|\boldsymbol{x}_0) = \boldsymbol{x}_1 - \boldsymbol{x}_0$.

Without loss of generality, the problem formulation could be extended to a flow defined in latent space. Accordingly, we will use $\boldsymbol{x}_t$ to denote both images and latent codes.

## 3 FLOWALIGN

### 3.1 TEXT-BASED IMAGE EDITING USING PRE-TRAINED FLOW MODELS

In text-based image editing using flow models, we aim to translate a source image $\boldsymbol{x}_{src}$ at $t = 1$ to a target image $\boldsymbol{x}_{tgt}$ at $t = 0$ based on text description of each image or editing instruction. In particular, such translation can be represented through a linear conditional flow between two image distributions,

$$\psi_t(\boldsymbol{x}_{src}|\boldsymbol{x}_{tgt}) := \boldsymbol{x}_t = (1 - t)\boldsymbol{x}_{tgt} + t\boldsymbol{x}_{src}. \tag{7}$$

Note that the associated flow ODE described by Eq. (5) requires training between two image distributions using the conditional flow matching of Eq. (3) with Eq. (7). To bypass the additional training, consider two generative flows that transfers the same noise $\boldsymbol{\epsilon} \sim \mathcal{N}(0, \mathbf{I}_d)$ to each image,

$$\psi_t^{src}(\boldsymbol{\epsilon}|\boldsymbol{x}_{src}) := \boldsymbol{q}_t = (1 - t)\boldsymbol{x}_{src} + t\boldsymbol{\epsilon}, \tag{8}$$

$$\psi_t^{tgt}(\boldsymbol{\epsilon}|\boldsymbol{x}_{tgt}) := \boldsymbol{p}_t = (1 - t)\boldsymbol{x}_{tgt} + t\boldsymbol{\epsilon}, \tag{9}$$

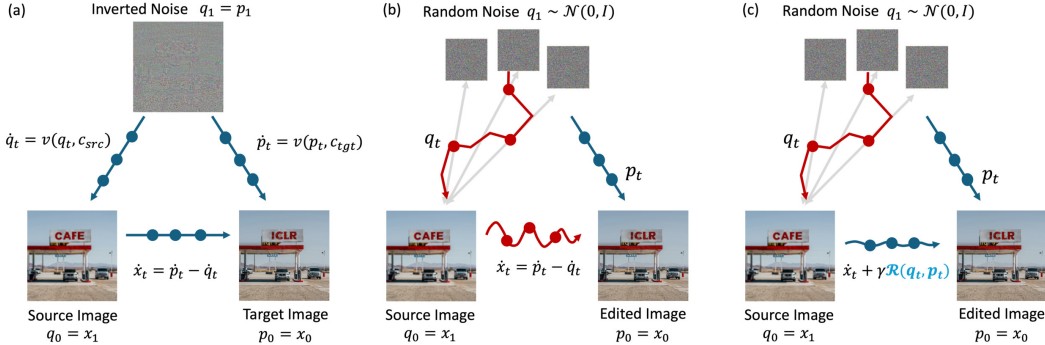

Figure 2: **Overview**. (a) Starting from the inverted latent, the ODE from source to target images can be obtained. (b) In contrast, existing inversion-free approaches suffer from nonsmooth trajectories, as $q_t$ is sampled with random noise at each step, often resulting in editing artifacts. (c) FlowAlign uses the regularized velocity $v_t^x$ from the similarity regularization at the terminal point ($\mathcal{R}$), producing smoother and more consistent trajectories between the source and target images. Prompt: *"...white and red sign that reads CAFE"* → *"...white and red sign that reads ICLR"*.

and the associated ODEs:

$$dq_t = v_t^{src}(q_t)dt, \quad dp_t = v_t^{tgt}(p_t)dt \tag{10}$$

with $q_1 = p_1 = \epsilon$, $q_0 = x_{src}$, and $p_0 = x_{tgt}$. Since both flows are accessible without additional training by using a pre-trained noise-to-image flow model[1], it is beneficial if we can leverage Eq. (8) and Eq. (9) to simulate the ODE for the flow in Eq. (7).

Importantly, the system of equations (Eq. (7), Eq. (8) and Eq. (9)) lead to the following key equality:

$$x_t = p_t - q_t + x_{src}, \quad \text{where} \quad x_1 = x_{src}, x_0 = x_{tgt} \tag{11}$$

Consequently, we can simulate the ODE for image editing by

$$dx_t = dp_t - dq_t = [v_t^{tgt}(p_t) - v_t^{src}(q_t)]dt, \quad \text{where} \quad p_t := q_t + x_t - x_{src} \tag{12}$$

without additional training of flow models (see Figure 2(a)). In case of text-conditional pre-trained flow models, we can leverage the same neural network $v_t^\theta$ with text embeddings $c_{src}$ and $c_{tgt}$ to approximate the $v_t^{tgt}(p_t), v_t^{src}(q_t)$, respectively, leading to

$$dx_t = [v_t^\theta(p_t, c_{tgt}) - v_t^\theta(q_t, c_{src})]dt. \tag{13}$$

## 3.2 TRAJECTORY ERRORS FROM INVERSION-FREE APPROACHES

In Eq. (13), $p_t$ and $q_t$ are first computed at each discrete time step $t$. If an inverted latent $\epsilon_{inv}$ is obtained via ODE inversion from the source image $x_{src}$ along the trajectory of $q_t$ (as illustrated in Figure 2(a)), then $p_t$ and $q_t$ can be further updated as:

$$dq_t = v_t^\theta(q_t, c_{src})dt, \quad dp_t = v_t^\theta(p_t, c_{tgt})dt \tag{14}$$

with the initialization condition $p_1 = q_1 = \epsilon_{inv}$. However, ODE inversion increase the overall computational cost. Thus, we are interested in avoiding the ODE inversion process. Toward this aim, FlowEdit (Kulikov et al., 2024) proposed the following update for $p_t$ and $q_t$

$$q_t = (1-t)x_{src} + t\epsilon, \quad dp_t = v_t^\theta(q_t + x_t - x_{src}, c_{tgt})dt \tag{15}$$

with the initialization condition $p_1 = q_1 = \epsilon$ for randomly sampled $\epsilon$ at each $t$, where the second equality stems from Eq. (11), i.e. $p_t = q_t + x_t - x_{src}$.

Unfortunately, one of the critical limitations of FlowEdit using Eq. (15) is that $q_t$ trajectory is not sufficiently smooth due to the random sampling of $\epsilon$ (Figure 2(b)). This leads to the inaccuracy of using simulated ODE in Eq. (13) for $x_t$. To mitigate this, we propose an explicit trajectory regularization method using the structural similarity at the terminal point.

---

[1] In this paper, we use Stable Diffusion 3.0 (medium), a foundational pre-trained flow model for the main experiments.

### 3.3 Trajectory Regularization using Similarity at Terminal Point

Given that the trajectory of $\boldsymbol{x}_t$ in Eq. (13) is not necessarily smooth due to the random sampling of $\boldsymbol{\epsilon}$, our objective is to simulate an ODE that explicitly penalizes deviations from a smooth trajectory as a form of regularization. Inspired by the recent advances of the optimal control approaches for flow models(Rout et al., 2025a;b), we consider the following time-reversal optimal control problem

$$\dot{\boldsymbol{x}}_t = \boldsymbol{u}(\boldsymbol{x}_t), \quad \boldsymbol{x}_1 = \boldsymbol{x}_{src} \tag{16}$$

$$V(\boldsymbol{u}_t) = \int_0^1 \ell(\boldsymbol{x}_t, \boldsymbol{u}_t, t)dt + m(\boldsymbol{x}_0) \tag{17}$$

In this paper, we use the following loss:

$$\ell(\boldsymbol{x}_t, \boldsymbol{u}_t, t) := \frac{1}{2}\|\boldsymbol{u}_t - \left(v_t^\theta(\boldsymbol{p}_t, c_{tgt}) - \boldsymbol{v}_t^\theta(\boldsymbol{q}_t, c_{src})\right)\|^2, \tag{18}$$

thereby enforcing the similarity of the optimal control $\boldsymbol{u}_t$ to the original velocity field in Eq. (13). Unfortunately, this is shown not sufficient for regularizing the trajectory deviation owing to the lack of inverted latent. Therefore, our goal is to utilize the terminal loss $m(\boldsymbol{x}_0)$ to enforce the trajectory smoothness.

One of the most important contributions of this work is showing that a similarity regularization at the terminal point serves the goal. Specifically, we introduce a $l_2$-based terminal point regularization:

$$\boldsymbol{m}(\boldsymbol{x}_0) = \frac{\eta}{2}\|\boldsymbol{x}_0 - \boldsymbol{x}_{src}\|^2 \tag{19}$$

This implies that the ODE evolution from from $t = 1$ to $t = 0$ ultimately converges to a terminal solution that closely resembles the starting point, i.e. $\boldsymbol{x}_{tgt} \simeq \boldsymbol{x}_{src}$. While this might raise concerns about convergence to a trivial solution where $\boldsymbol{x}_t = \boldsymbol{x}_{src}$ for all $t$, the crucial distinction lies in our use of a finite $\eta$, as opposed to taking the limit $\eta \to \infty$ as done in RF-inversion (Rout et al., 2025a). In what follows, we show that this terminal point regularization lead to the balance between the semantic guidance and structural preservation.

**Proposition 1.** *For the linear conditional flow with $a_t = 1 - t$ and $b_t = t$ where $0 \leq t \leq 1$, the ODE that solves the optimal control problem with Eq. (17), Eq. (18) and Eq. (19) is given by*

$$d\boldsymbol{x}_t = \boldsymbol{v}_t^{\boldsymbol{x}}(\boldsymbol{p}_t, \boldsymbol{q}_t, \boldsymbol{p}_0, \boldsymbol{q}_0)dt \tag{20}$$

*with the initial condition $\boldsymbol{x}_1 = \boldsymbol{x}_{src}$, where*

$$\boldsymbol{v}_t^{\boldsymbol{x}}(\boldsymbol{p}_t, \boldsymbol{q}_t, \boldsymbol{p}_0, \boldsymbol{q}_0) :\simeq \boldsymbol{v}_t(\boldsymbol{p}_t, c_{tgt}) - \boldsymbol{v}_t(\boldsymbol{q}_t, c_{src}) + \gamma\left(\mathbb{E}[\boldsymbol{p}_0|\boldsymbol{p}_t] - \mathbb{E}[\boldsymbol{q}_0|\boldsymbol{q}_t]\right) \tag{21}$$

*where $\gamma = \frac{\eta}{\eta t - 1}$ is positive for sufficiently large $\eta$ and $\mathbb{E}[\boldsymbol{q}_0|\boldsymbol{q}_t] = \boldsymbol{q}_t - t\boldsymbol{v}_t(\boldsymbol{q}_t, c_{src}), \mathbb{E}[\boldsymbol{p}_0|\boldsymbol{p}_t] = \boldsymbol{p}_t - t\boldsymbol{v}_t(\boldsymbol{p}_t, c_{tgt})$ are Tweedie's denoising estimates.*

Note that the resulting velocity field can be decomposed as

$$\boldsymbol{v}_t^{\boldsymbol{x}}(\boldsymbol{p}_t, \boldsymbol{q}_t, \boldsymbol{p}_0, \boldsymbol{q}_0) := \underbrace{[\boldsymbol{v}_t(\boldsymbol{p}_t, c_{tgt}) - \boldsymbol{v}_t(\boldsymbol{q}_t, c_{src})]}_{\text{Semantic Guidance}} + \gamma\underbrace{\left(\mathbb{E}[\boldsymbol{p}_0|\boldsymbol{p}_t] - \mathbb{E}[\boldsymbol{q}_0|\boldsymbol{q}_t]\right)}_{\text{Source Consistency}(\mathcal{R})}. \tag{22}$$

The final velocity in Eq. (22) needs more discussion. Similar to the drift term in Eq. (13), the first term in Eq. (22) serves as the directional signal from the source to the target semantics. On the other hand, the second term becomes a source consistent regularization gradient based on the distance between the clean estimates of $\boldsymbol{p}_t$ and $\boldsymbol{q}_t$, computed using Tweedie formula. This regularization gradient keeps the trajectory $\boldsymbol{p}_t$ close to the source-consistency direction represented by $\mathbb{E}[\boldsymbol{q}_0|\boldsymbol{q}_t]$, thereby implicitly regulating $\boldsymbol{x}_t$.

Another byproduct of trajectory regularization is the robustness to the classifier-free guidance (CFG). Unlike recent work (Kulikov et al., 2024) that uses CFG to both estimated noises in with null-text embedding and different scales, we use the CFG as for the ODE trajectory $\boldsymbol{p}_t$ only,

$$\boldsymbol{v}^\theta(\boldsymbol{p}_t, c_{src}, c_{tgt}) = \boldsymbol{v}^\theta(\boldsymbol{p}_t, c_{src}) + \omega\left[\boldsymbol{v}^\theta(\boldsymbol{p}_t, c_{tgt}) - \boldsymbol{v}^\theta(\boldsymbol{p}_t, c_{src})\right] \tag{23}$$

where $\omega \geq 0$ is the CFG scale factor. We find that commonly used CFG values (such as 7.5) are effective. For further analysis of the effect of $\omega$, see Section 4. Accordingly, the proposed method is presented as time-reversal regularized trajectory:

$$d\boldsymbol{x}_t = [\boldsymbol{v}_t(\boldsymbol{p}_t, c_{tgt}, c_{src}) - \boldsymbol{v}_t(\boldsymbol{q}_t, c_{src})]dt - \gamma dt(\mathbb{E}[\boldsymbol{q}_0|\boldsymbol{q}_t] - \mathbb{E}[\boldsymbol{p}_0|\boldsymbol{p}_t]), \quad \boldsymbol{x}_1 = \boldsymbol{x}_{src} \tag{24}$$

The complete algorithm is presented in Algorithm 1. Our approach introduces two hyperparameters $\omega$ and $\zeta = -\gamma dt > 0$. We find that using constant values yields stable results and provide detailed analysis of their effects in Section 4.

---

**Algorithm 1** Algorithm of FlowAlign on Latent Space

---

**Require:** Source image $\boldsymbol{z}_{src}$, Pre-trained flow model $\boldsymbol{v}^\theta$, VAE encoder and Decoder $\mathcal{E}, \mathcal{D}$, Source/Target text embeddings $c_{src}, c_{tgt}$, CFG scale $\omega$, source consistency scale $\zeta$

1: $\boldsymbol{x}_{src} \leftarrow \mathcal{E}(\boldsymbol{z}_{src})$
2: $\boldsymbol{x}_t \leftarrow \boldsymbol{x}_{src}$
3: **for** $t : 1 \rightarrow 0$ **do**
4: $\quad \epsilon \sim \mathcal{N}(0, \mathbf{I})$
5: $\quad \boldsymbol{q}_t \leftarrow (1 - t)\boldsymbol{x}_{src} + t\boldsymbol{\epsilon}$
6: $\quad \boldsymbol{p}_t \leftarrow \boldsymbol{x}_t - \boldsymbol{x}_{src} + \boldsymbol{q}_t$
7: $\quad \boldsymbol{v}^\theta(\boldsymbol{p}_t) := \boldsymbol{v}^\theta(\boldsymbol{p}_t, c_{src}) + \omega \left[ \boldsymbol{v}^\theta(\boldsymbol{p}_t, c_{tgt}) - \boldsymbol{v}^\theta(\boldsymbol{p}_t, c_{src}) \right]$
8: $\quad \boldsymbol{v}^\theta(\boldsymbol{q}_t) := \boldsymbol{v}^\theta(\boldsymbol{q}_t, c_{src})$
9: $\quad \mathbb{E}[\boldsymbol{p}_0 | \boldsymbol{p}_t] \leftarrow \boldsymbol{p}_t - t\boldsymbol{v}^\theta(\boldsymbol{p}_t), \quad \mathbb{E}[\boldsymbol{q}_0 | \boldsymbol{q}_t] \leftarrow \boldsymbol{q}_t - t\boldsymbol{v}^\theta(\boldsymbol{q}_t)$
10: $\quad \boldsymbol{x}_t \leftarrow \boldsymbol{x}_t + \left[ \boldsymbol{v}^\theta(\boldsymbol{p}_t) - \boldsymbol{v}^\theta(\boldsymbol{q}_t) \right] dt + \zeta(\mathbb{E}[\boldsymbol{q}_0 | \boldsymbol{q}_t] - \mathbb{E}[\boldsymbol{p}_0 | \boldsymbol{p}_t])$
11: **end for**
12: $\boldsymbol{z}_{edit} \leftarrow \mathcal{D}(\boldsymbol{x}_t)$

---

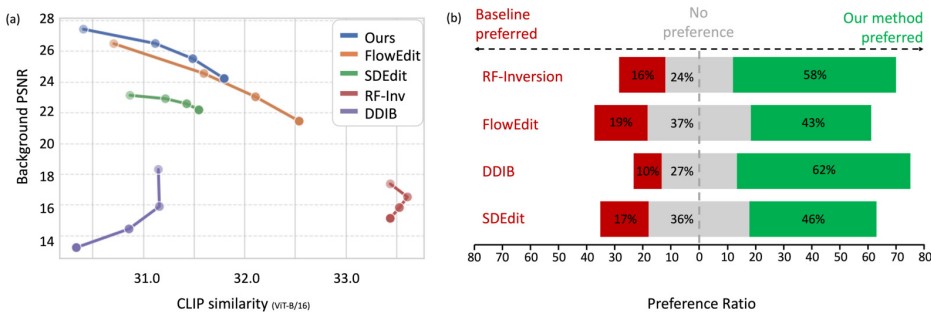

Figure 3: **Quantitative and human preference evaluation**. (a) Trade-off between CLIP similarity versus background PSNR. The darker the blob, the larger CFG scale. (b) User preference results. Each bar shows the proportion of responses favoring the baseline (red), showing no preference (gray), or favoring our method (green). Bars are centered at 0 to emphasize directional preference. Across all comparisons, our method is preferred.

## 4 EXPERIMENTAL RESULTS

**Dataset, Baseline and Compute Resource.** To verify the editing performance, we perform multiple analysis using PIE-Bench (Ju et al., 2023) that contains 700 synthetic and natural images with paired original and editing prompts. For baseline algorithms, we focus on comparing with methods that establish trajectory between two samples. For the methods that utilizes noisy sample distribution to connect two trajectories, we use SDEdit (Meng et al., 2021) and DDIB (Su et al., 2023). For the method that improves the inversion process, we use RF-inversion (Rout et al., 2025a). For the inversion-free method, we select FlowEdit (Kulikov et al., 2024). Note that all these methods are training-free text-based image editing algorithms. For a fair comparison, we use the same flow model and 33 NFEs by following FlowEdit. In case of methods that involves inversion process, we use 17 NFEs for each inversion and sampling process. We conduct experiments using NVIDIA GeForce RTX 4090 (24GB VRAM). For runtime analysis, see Appendix D.

**Semantic alignment and Structure Consistency.** Image editing quality should be evaluated from two complementary perspectives: semantic alignment and source structure consistency. An effective editing method should achieve a balance between these two objectives, improving both simultaneously. To assess our method, we follow prior works and report CLIP similarity as a measure of semantic alignment and background PSNR as a proxy for structural consistency. We evaluate performance across various classifier-free-guidance (CFG) scales {5.0, 7.5, 10.0, 13.5}, as CFG is a key parameter that modulates the trade-off between semantic fidelity and structure consistency in flow-based, text-guided image editing. Figure 3a presents the quantitative evaluation results. For the complete results, please refer to Appendix F. While the trade-off between semantic alignment and structure consistency remains, the proposed method consistently achieves higher structural preservation compared to all other methods. In terms of semantic alignment, measured by CLIP similarity, the proposed method

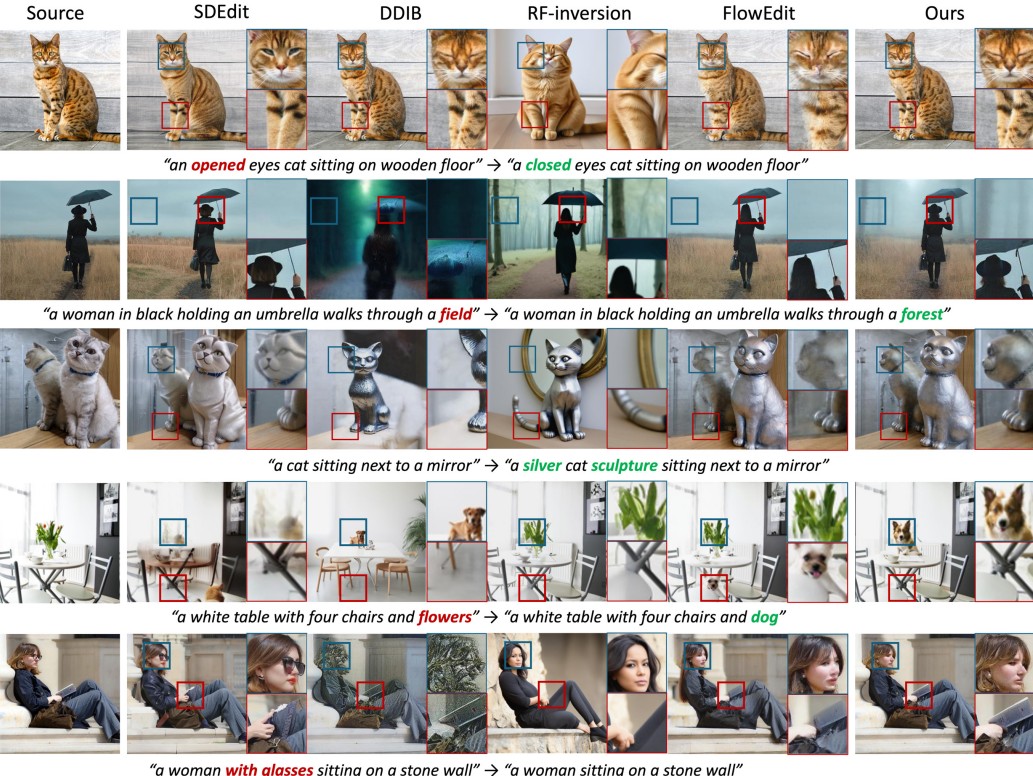

|Source|SDEdit|DDIB|RF-inversion|FlowEdit|Ours|

*"an opened eyes cat sitting on wooden floor"* → *"a closed eyes cat sitting on wooden floor"*

*"a woman in black holding an umbrella walks through a field"* → *"a woman in black holding an umbrella walks through a forest"*

*"a cat sitting next to a mirror"* → *"a silver cat sculpture sitting next to a mirror"*

*"a white table with four chairs and flowers"* → *"a white table with four chairs and dog"*

*"a woman with glasses sitting on a stone wall"* → *"a woman sitting on a stone wall"*

Figure 4: Qualitative comparison of text-based image editing methods. Insets provide zoomed-in views of regions highlighted by red and blue rectangles. Our method achieves better semantic alignment and structure consistency across a diverse set of prompts.

outperforms SDEdit and DDIB, while FlowEdit and RF-Inversion achieves higher scores. However, these baselines tend to increase CLIP similarity at the cost of source structure consistency. In many cases, their high CLIP scores result from over-expression of target prompt objects, often distorting the original image as shown in Figure 4.

**Human evaluation.** Due to limitations of current metrics, such as CLIP similarity (as discussed in the previous section), we additionally conduct a human preference study. From a pool of 700 validation samples, we randomly select 100 images along with their corresponding original and editing text prompts. For each participant, we present a pairwise comparison between the edited image produced by the proposed method and that of a randomly selected baseline. Participants are asked to indicate which result they prefer, that is defined as one with accurate reflectance of editing prompt with source structure preservation. For a more detailed description of the human evaluation protocol, refer to Appendix C. Figure 3b summarizes the preference ratio of out method against each baseline. Across all comparisons, the proposed method is more preferred. These results demonstrates that our method achieves improved editing fidelity and superior source structure consistency, as intended.

Table 1: Quantitative comparison for backward editing results. **Bold** represents the best.

| Metric | PSNR ↑ | DINO Dist ↓ | LPIPS ↓ | MSE ↓ |
|---|---|---|---|---|
| SDEdit (Meng et al., 2021) | 13.83 | 0.078 | 0.419 | 0.043 |
| DDIB (Su et al., 2023) | 18.18 | 0.041 | 0.190 | 0.019 |
| RF-Inv (Rout et al., 2025a) | 12.14 | 0.113 | 0.502 | 0.065 |
| FlowEdit (Kulikov et al., 2024) | 19.88 | 0.037 | 0.147 | 0.012 |
| Ours | **27.42** | **0.025** | **0.085** | **0.006** |

**Backward Editing.** In this work, we proposed a flow matching regularization to make smoother trajectory between two image samples. Thus, we investigate whether the proposed trajectory Eq. (24) behaves deterministically between two samples. To test this, we solve Eq. (24) in backward direction starting from the edited image. By evaluating the similarity between the source image and recon-

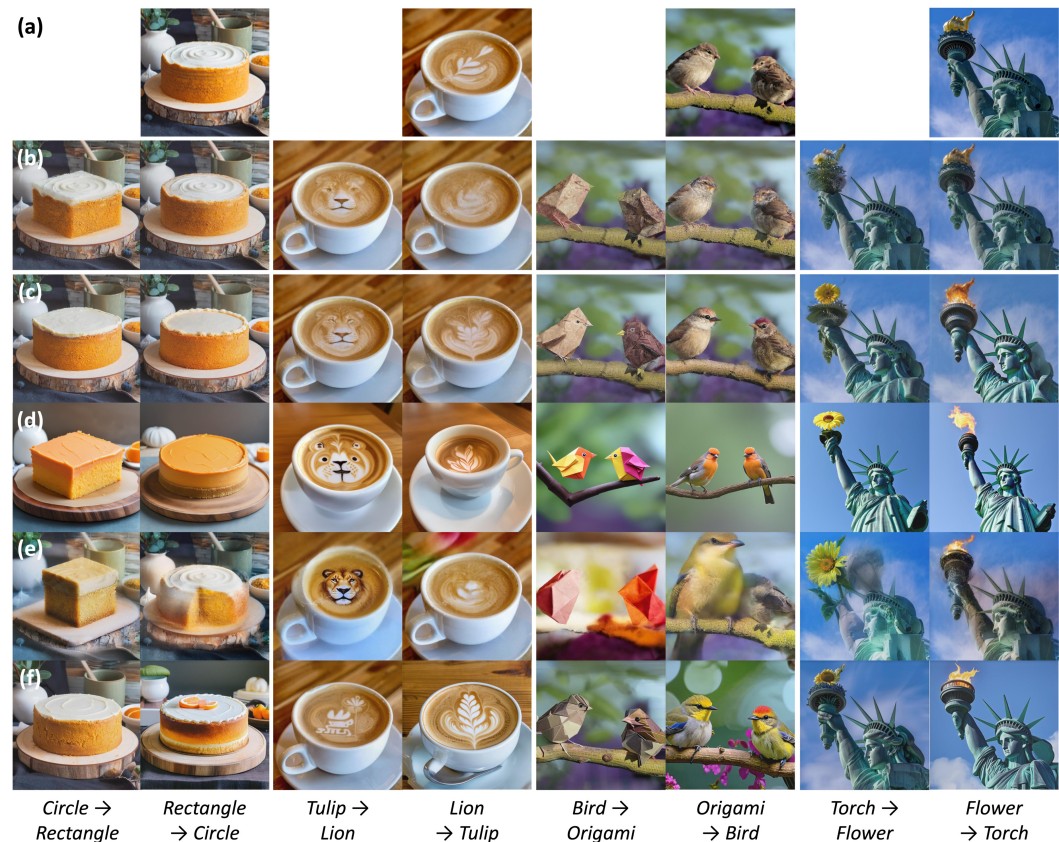

Figure 5: Qualitative comparison of editing (odd columns) and backward editing (even columns) results. FlowAlign (the 2nd row) achieves better reconstruction quality compared to the baselines. (a) Source image, (b) FlowAlign, (c) FlowEdit, (d) RF-inversion, (e) DDIB, and (f) SDEdit.

structed image, we can evaluate whether the editing method indeed shows behavior like an ODE. We compute both pixel-wise metrics (PSNR, MSE) and perceptual metrics (LPIPS, DINO structural distance) and report the results in Table 1. Across all metrics, the proposed method outperforms the baselines. These results suggest that reverse editing using the proposed method nearly reconstructs the source image, supporting the effectiveness of the proposed trajectory regularization. Figure 5 shows a qualitative comparison for the edited image and reconstructed image. For example, in the last column, the proposed method uniquely reconstruct the torch of the statue to match the source image, whereas the baselines fail to reconstruct it accurately and instead convert it into a real torch. Importantly, the edited image obtained by the proposed method faithfully reflects the intended editing direction. This implies that the observed reconstruction ability stems from the smooth trajectory, rather than simply reducing changes during the editing process.

**Ablation Study.** The proposed method introduces two hyperparameters, $\omega$ and $\zeta$, which control the relative strength of each regularization term in Eq. (22). These parameters govern the trade-off between semantic alignment with the target prompt and structural consistency with the source image. To evaluate their impact, we conduct an ablation study under various settings. As shown in Figure 6, we observe that setting $\zeta = 0.01$ consistently achieves a favorable balance between semantic alignment and structure preservation. Notably, the framework without any additional gradients (i.e. $\omega = \zeta = 0$, block dot) results in insufficient edit-

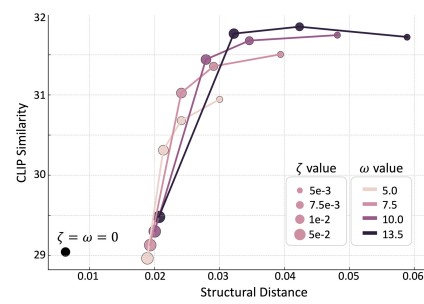

Figure 6: Ablation study for $\omega$ and $\zeta$. Top-left points represent to balanced performance.

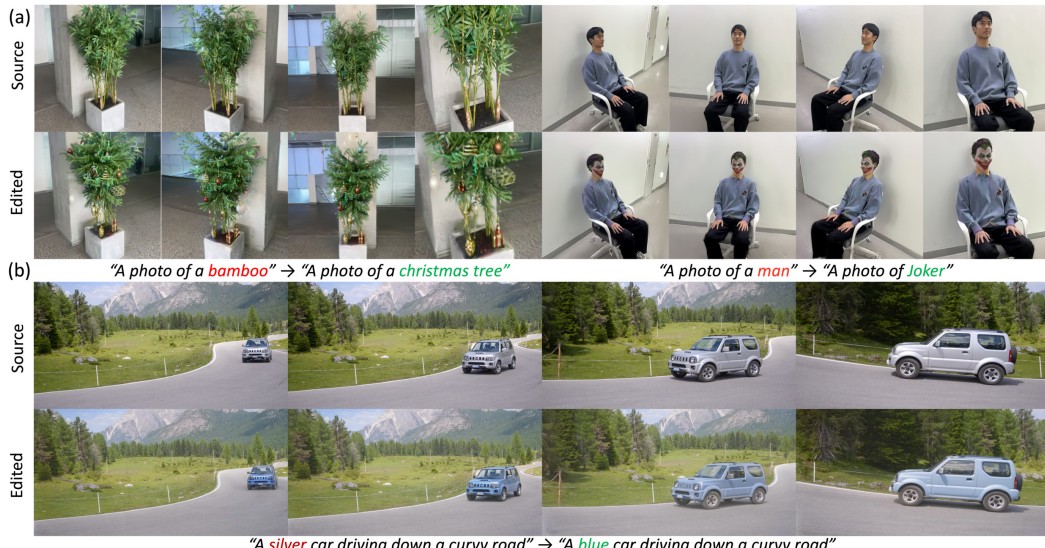

Figure 7: Editing results for (a) 3D Gaussian splatting rendered from four different viewpoints, and (b) a video sequence edited frame-by-frame.

ing, which appears at the left-bottom position. This result emphasizes the effectiveness of the proposed CFG.

**Further Applications.** Figure 7a demonstrates the application of FlowAlign to 3D editing via Gaussian splatting, highlighting its effectiveness in editing Gaussian parameters using FlowAlign in place of standard score distillation—thus extending its utility beyond 2D image editing. Figure 7b illustrates the application of FlowAlign to video editing, where it is applied independently to each frame. Although temporal consistency is not explicitly enforced, the strong source consistency of FlowAlign results in visually coherent backgrounds across frames. Additional details and results for these applications are provided in Appendix G.

## 5 CONCLUSION

In this paper, we propose a flow matching regularization, that leads to smooth and stable editing trajectory, for inversion-free flow-based image editing algorithm. By defining differentiable optimal control problem with similarity regularization at the terminal point, we explicitly balance semantic alignment with editing text and structural consistency with the source image. As a result, the proposed method achieves comparable or better editing performance while showing superior source consistency and better computational efficiency. Notably, we demonstrate that the simulated ODE exhibits a deterministic property by performing the reverse editing experiments. In summary, this work provides a novel design space for approximating ODEs between two samples without additional training.

**Limitation and Potential Negative Impacts.** While the proposed method offers efficient training-free image editing algorithm, it requires multiple diffusion timesteps for inference. The proposed method leverages noise-to-image flow-based models for image editing, it inherits potential shortcomings of the underlying noise-to-image flow-based models, such as biased generation. In addition, image editing techniques may be misused for harmful applications, and research addressing these societal risks should be considered alongside technical advances.

### ACKNOWLEDGMENTS

This work was supported by the National Research Foundation of Korea under Grant RS-2024-00336454, the Institute of Information & Communications Technology Planning & Evaluation(IITP) grant funded by the Korea government(MSIT) (RS-2025-02304967, AI Star Fellowship(KAIST); No. RS-2024-00457882, AI Research Hub Project), the AI Computing Infrastructure Enhancement (GPU Rental Support) User Support Program funded by the Ministry of Science and ICT (MSIT), Republic of Korea (RQT-25-120217).

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

## A  OPTIMAL CONTROL FORMULATION TO PROVE PROPOSITION 1

The following lemma is required for the proof.

**Lemma 1.** *Consider the following time-reversal optimal control problem:*

$$V(\boldsymbol{u}_t) = \int_0^1 \ell(\boldsymbol{x}_t, \boldsymbol{u}_t, t)dt + m(\boldsymbol{x}_0), \quad \dot{\boldsymbol{x}}_t = \boldsymbol{u}(\boldsymbol{x}_t), \quad \boldsymbol{x}_{t_0} = \boldsymbol{x}_{start}, \tag{25}$$

*where*

$$\ell(\boldsymbol{x}_t, \boldsymbol{u}_t, t) := \frac{1}{2}\|\boldsymbol{u}_t - \boldsymbol{a}_t\|^2 \tag{26}$$

$$m(\boldsymbol{x}_0) := \frac{\eta}{2}\|\boldsymbol{x}_0 - \boldsymbol{x}_{src}\|^2 \tag{27}$$

*Then, the optimal solution trajectory is given by*

$$\dot{\boldsymbol{x}}_t = -\boldsymbol{p}_t + \boldsymbol{a}_t \tag{28}$$

*where $\boldsymbol{p}_t$ is given by*

$$\boldsymbol{p}_t = \frac{\eta}{\eta t - 1}(\boldsymbol{x}_t + \boldsymbol{v}_t - \boldsymbol{x}_{src}) \quad where \quad \boldsymbol{v}_t := \int_t^0 \boldsymbol{a}_t dt \tag{29}$$

*Proof.* The Hamiltonian for the given optimal control problem can be represented by

$$H(\boldsymbol{x}_t, \boldsymbol{u}_t, \boldsymbol{p}_t, t) := \frac{1}{2}\|\boldsymbol{u}_t - \boldsymbol{a}_t\|^2 + \boldsymbol{p}_t^T \boldsymbol{u}_t \tag{30}$$

The optimal control $\boldsymbol{u}_t^*$ that minimizes the Hamitonian is then given by

$$\boldsymbol{u}_t^* = \boldsymbol{a}_t - \boldsymbol{p}_t \tag{31}$$

This leads to the following

$$H(\boldsymbol{x}_t, \boldsymbol{u}_t^*, \boldsymbol{p}_t, t) := -\frac{1}{2}\|\boldsymbol{p}_t\|^2 + \boldsymbol{p}_t^T \boldsymbol{a}_t \tag{32}$$

Then, the minimum principle (Basar et al., 2020; Fleming & Rishel, 2012) informs that the optimal pair $(\boldsymbol{x}_t, \boldsymbol{p}_t)$ should satisfy the following:

$$\dot{\boldsymbol{x}}_t = \boldsymbol{u}_t^* = \boldsymbol{a}_t - \boldsymbol{p}_t \tag{33}$$

$$\dot{\boldsymbol{p}}_t = -\frac{\partial H(\boldsymbol{x}_t, \boldsymbol{u}_t^*, \boldsymbol{p}_t, t)}{\partial \boldsymbol{x}} = \boldsymbol{0} \tag{34}$$

with the additional boundary conditions

$$\boldsymbol{x}_{t_0} = \boldsymbol{x}_{start}, \quad \boldsymbol{p}_0 = \frac{\partial m(\boldsymbol{x}_t, t)}{\partial \boldsymbol{x}_t}\bigg|_{t=0} = \eta(\boldsymbol{x}_0 - \boldsymbol{x}_{src}) \tag{35}$$

From Eq. (34), we can see that $\boldsymbol{p}_t$ is time-invariant constant, i.e. $\boldsymbol{p}_t = \boldsymbol{p}$ for all $t \in [0, 1]$. Accordingly, we have

$$\boldsymbol{x}_0 = \boldsymbol{x}_{t_0} + \int_{t_0}^0 -\boldsymbol{p} + \boldsymbol{a}_t dt \tag{36}$$

$$= t_0 \boldsymbol{p} + \boldsymbol{x}_{t_0} + \boldsymbol{v}_{t_0} \tag{37}$$

where

$$\boldsymbol{v}_{t_0} := \int_{t_0}^0 \boldsymbol{a}_t dt \tag{38}$$

By plugging this in Eq. (35) with $\boldsymbol{p}_0 = \boldsymbol{p}$, we have

$$\boldsymbol{p} = \eta(t_0\boldsymbol{p} + \boldsymbol{x}_{t_0} + \boldsymbol{v}_{t_0} - \boldsymbol{x}_{src}) \quad \Rightarrow \boldsymbol{p} = \frac{\eta(\boldsymbol{x}_{t_0} + \boldsymbol{v}_{t_0} - \boldsymbol{x}_{src})}{1 - t_0\eta}$$

Therefore, the optimal control $\boldsymbol{u}_t^*$ is given by

$$\boldsymbol{u}_t^* = \boldsymbol{a}_t + \frac{\eta}{\eta t - 1}(\boldsymbol{x}_t + \boldsymbol{v}_t - \boldsymbol{x}_{src}) \tag{39}$$

$\square$

Now we are ready to prove our main results.

**Proposition 1.** *For the linear conditional flow with $a_t = 1 - t$ and $b_t = t$ where $0 \leq t \leq 1$, the ODE that solves the optimal control problem with Eq. (17), Eq. (18) and Eq. (19) is given by*

$$d\boldsymbol{x}_t = \boldsymbol{v}_t^{\boldsymbol{x}}(\boldsymbol{p}_t, \boldsymbol{q}_t, \boldsymbol{p}_0, \boldsymbol{q}_0)dt \tag{20}$$

*with the initial condition $\boldsymbol{x}_1 = \boldsymbol{x}_{src}$, where*

$$\boldsymbol{v}_t^{\boldsymbol{x}}(\boldsymbol{p}_t, \boldsymbol{q}_t, \boldsymbol{p}_0, \boldsymbol{q}_0) :\simeq \boldsymbol{v}_t(\boldsymbol{p}_t, c_{tgt}) - \boldsymbol{v}_t(\boldsymbol{q}_t, c_{src}) + \gamma \left(\mathbb{E}[\boldsymbol{p}_0|\boldsymbol{p}_t] - \mathbb{E}[\boldsymbol{q}_0|\boldsymbol{q}_t]\right) \tag{21}$$

*where $\gamma = \frac{\eta}{\eta t - 1}$ is positive for sufficiently large $\eta$ and $\mathbb{E}[\boldsymbol{q}_0|\boldsymbol{q}_t] = \boldsymbol{q}_t - t\boldsymbol{v}_t(\boldsymbol{q}_t, c_{src}), \mathbb{E}[\boldsymbol{p}_0|\boldsymbol{p}_t] = \boldsymbol{p}_t - t\boldsymbol{v}_t(\boldsymbol{p}_t, c_{tgt})$ are Tweedie's denoising estimates.*

*Proof.* We can now use Lemma 1 with the following modification

$$\boldsymbol{a}_t := \boldsymbol{v}_t^\theta(\boldsymbol{p}_t, c_{tgt}) - \boldsymbol{v}_t^\theta(\boldsymbol{q}_t, c_{src}) \tag{40}$$

This leads to the following first order approximation:

$$\boldsymbol{v}_t = \int_t^0 \boldsymbol{a}_t dt = \int_t^0 \boldsymbol{v}_t(\boldsymbol{p}_t) - \boldsymbol{v}_t(\boldsymbol{q}_t)dt \tag{41}$$

$$\simeq -t\boldsymbol{v}_t(\boldsymbol{p}_t) + t\boldsymbol{v}_t(\boldsymbol{q}_t) \tag{42}$$

Accordingly, we have

$$\begin{aligned} \boldsymbol{x}_t + \boldsymbol{v}_t - \boldsymbol{x}_{src} &= \boldsymbol{p}_t - \boldsymbol{q}_t + \boldsymbol{v}_t \\ &\simeq \boldsymbol{p}_t - \boldsymbol{q}_t + (-t\boldsymbol{v}_t(\boldsymbol{p}_t) + t\boldsymbol{v}_t(\boldsymbol{q}_t)) \\ &= \mathbb{E}[\boldsymbol{p}_0|\boldsymbol{p}_t] - \mathbb{E}[\boldsymbol{q}_0|\boldsymbol{q}_t] \end{aligned}$$

where we use $\boldsymbol{x}_t - \boldsymbol{x}_{src} = \boldsymbol{p}_t - \boldsymbol{q}_t$ from Eq. (11) for the first equality. Thus, the optimal control $\boldsymbol{u}_t^*$ is given by

$$\begin{aligned} \boldsymbol{u}_t^* &= \boldsymbol{a}_t + \frac{\eta}{\eta t - 1}\left(\boldsymbol{x}_t + \boldsymbol{v}_t - \boldsymbol{x}_{src}\right) \\ &\simeq \boldsymbol{a}_t + \frac{\eta}{\eta t - 1}\left(\mathbb{E}[\boldsymbol{p}_0|\boldsymbol{p}_t] - \mathbb{E}[\boldsymbol{q}_0|\boldsymbol{q}_t]\right) \end{aligned} \tag{43}$$

Therefore, we have

$$\dot{\boldsymbol{x}}_t = \boldsymbol{v}_t(\boldsymbol{p}_t, c_{tgt}) - \boldsymbol{v}_t(\boldsymbol{q}_t, c_{src}) + \frac{\eta}{\eta t - 1}\left(\mathbb{E}[\boldsymbol{p}_0|\boldsymbol{p}_t] - \mathbb{E}[\boldsymbol{q}_0|\boldsymbol{q}_t]\right) \tag{44}$$

$$= \boldsymbol{v}_t(\boldsymbol{p}_t, c_{tgt}) - \boldsymbol{v}_t(\boldsymbol{q}_t, c_{src}) + \gamma \left(\mathbb{E}[\boldsymbol{p}_0|\boldsymbol{p}_t] - \mathbb{E}[\boldsymbol{q}_0|\boldsymbol{q}_t]\right) \tag{45}$$

$\square$

# B    IMPLEMENTATION DETAILS

In this section, we provide implementation details of baseline methods and the proposed method. For the proposed method, we will release the code to `https://github.com/FlowAlign/FlowAlign`.

**Backbone model**    While various text-to-image generative models exist, we leverage a flow-based model defined in latent space, specifically using Stable Diffusion 3.0 (medium) (Esser et al., 2024) provided by the diffusers package. Given the significantly improved generative and text alignment performance of this backbone compared to earlier versions, we use the same backbone model for all baselines to establish a fair comparison. For the time discretizations, we set the shift coefficient to 3.0, which is a default option of the Stable Diffusion 3.0

**Baseline methods**

1. DDIB (Su et al., 2023) : DDIB involves an inversion process followed by a sampling process. For inversion, we adopt the backward flow ODE. Regarding classifier-free guidance (CFG), we use only the null-text embedding during inversion and both target text and null-text embeddings during the sampling. This choice is motivated by the instability observed when applying standard CFG (i.e., using the source text and null-text embeddings) during inversion. We set the number of ODE timesteps to 17 for both the inversion and sampling processes, to conduct a comparison under similar number of total ODE timesteps.

2. SDEdit (Meng et al., 2021) : SDEdit requires specifying the initial SNR (i.e. timestep), and its performance can vary significantly depending on this choice. In this work, our main focus is to address the limitations of inversion-free editing methods. To fairly demonstrate the effectiveness of our approach in comparison to alternative methods that also can mitigate this issue, we adopt the same initial SNR setting as FlowEdit (Kulikov et al., 2024), determined by the starting timestep of the ODE. Specifically, we use the 18th timestep as the starting point in our experiments.

3. RF-inversion (Rout et al., 2025a) : RF-Inversion introduces an optimal-control-based guidance mechanism that ensures the inverted representation aligns with a target terminal state, resuling in a sampling process that is more likely under a predefined terminal distribution. We follow the official implementation, setting $\gamma = 0.5$, $\eta = 0.9$, the starting time $s = 0$, and the stopping time $\tau = 0.25$. RF-Inversion uses only the null-text embedding during inversion, while both the target text and null-text embeddings are used during the sampling phase.

4. FlowEdit (Kulikov et al., 2024) : We follow the official implementation of FlowEdit, setting the CFG scale to 3.5 for the source direction. Additionally, we solve the flow ODE starting from the 18th timestep out of 50, resulting in 33 ODE timesteps.

**Evaluation Metrics**   For the quantitative comparison, we evaluate following metrics using the official evaluation code [2] from PIEBench (Ju et al., 2024):

1. Background PSNR : PIEbench (Ju et al., 2024) provides masks that cover the object to be edited. Accordingly, we compute the PSNR by excluding the masked region, resulting in the background PSNR.

2. Background LPIPS : We measure the LPIPS (Blau & Michaeli, 2018), which is defined as distance between feature maps of pre-trained VGG network, by excluding the masked region.

3. Background SSIM : We compute the structural similarity (Wang et al., 2004) by excluding the masked region.

4. Background MSE : we compute pixel-wise mean-squared-error by excluding the masked region.

5. CLIP-score : We report the similarity between features embedded by pre-trained CLIP (Radford et al., 2021)[3] image encoder and text encoder.

6. HPS : We measure the Human Preference Score (HPS) (Wu et al., 2023), a metric trained on a large-scale dataset of human feedback to predict the aesthetic quality and text-image alignment.

## C   HUMAN PREFERENCE TEST PROTOCOL

To evaluate the quality of image editing, we conduct a human preference study in the form of an AB-test. Specifically, we randomly sample 100 images from the 700 validation samples of PIEBench (Ju et al., 2024). The study follows the protocol below for each participant:

1. Randomly select one sample from the 100-image pool.

---

[2] https://github.com/cure-lab/PnPInversion/tree/main/evaluation
[3] We use CLIP ViT-base-patch16 and ViT-large-patch14.

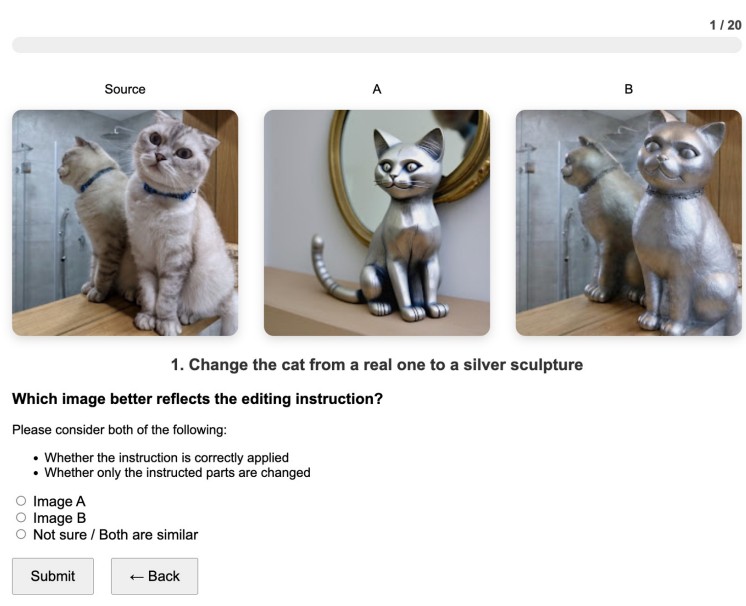

Figure 8: User interface for AB-test.

2. Randomly choose one baseline method from the four.

3. Randomly assign the baseline and the proposed method to gruops A and B.

4. Display AB-test user instruction with editing instruction, source image, and edited results from both methods.

5. The participant selects one of the following options: "A is better", "B is better", or "Not sure".

6. The participant clicks "Submit", and the response is recorded.

7. Step 1-6 are repeated until 20 cases are completed.

8. If more than half of the responses are "Not sure", an additional 5 comparisons is presented following the same protocol.

Although the editing methods utilize source–target text pairs, we present the editing instructions from PIEBench to participants instead, aiming to improve readability and reduce cognitive load. The AB-test interface is shown Fig. 8, and was implement using Google Script. We recruited 25 participants and collected a total of 505 responses (with one participant receiving additional cases due to frequent "Not sure" selections). These responses were used to compute the final human preference results.

## D    RUN-TIME COMPARISON

Table 2: Runtime comparison (unit: seconds). Averaged time for 50 samples is reported.

|  | DDIB | SDEdit | RF-inv | FlowEdit | FlowAlign |
|---|---|---|---|---|---|
| Stable Diffusion 3.0 | 4.06 | 5.26 | 4.06 | 10.64 | 7.14 |

In this section, we report the wall-clock time for each editing method evaluated in the main paper, specifically focusing on the runtime for solving ODE in latent space. We use a single RTX 4090 to measure the runtime. All experiments are conducted with the similar number of ODE timesteps where the sample $x_t$ is updated. For inversion-based methods, we update sample with 17 ODE timesteps for inversion and 17 ODE timesteps for the sampling, and for inversion-free methods, ODE timesteps

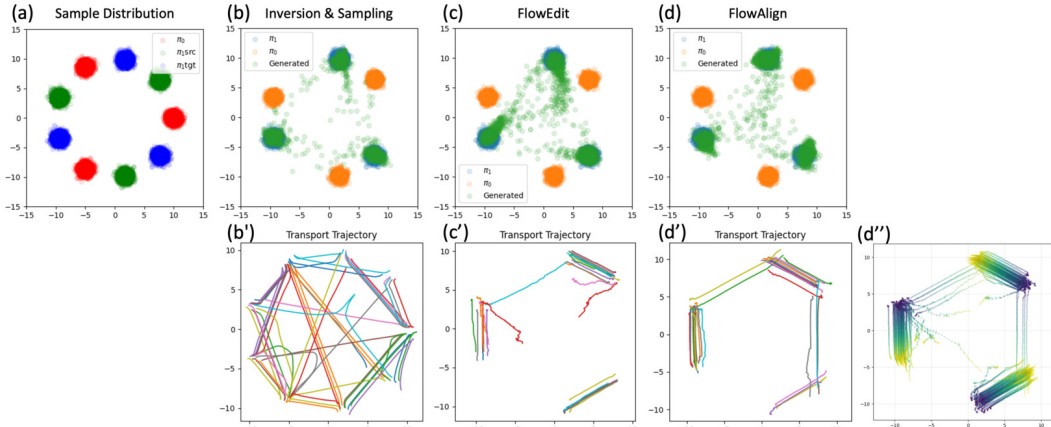

Figure 9: Results of inversion-based transformation, FlowEdit and FlowAlign on 2D toy dataset.

are set to 33. Table 2 shows the averaged runtime (in seconds) measured across 50 editing cases. For SDEdit, the runtime corresponds to standard reverse sampling with CFG. In comparison, DDIB and RF inversion achieve slightly shorter runtime, as we set the CFG scale to 0 during inversion. This is because applying CFG in the inversion stage often amplifies errors and disrupts the sampling process. FlowEdit requires computing CFG twice per ODE timestep - once for the source direction and once for the target - which results in approximately double the runtime of SDEdit. In other words, when using the same discretized ODE schedule, FlowEdit incurs a similar computational cost to inversion-based methods. In contrast, the proposed method achieves faster runtime compared to FlowEdit due to its efficient CFG, which is computed only for $v(p_t)$. Importantly, this computational efficiency is achieved without sacrificing performance.

# E  ANALYSIS ABOUT EDITING TRAJECTORY

To provide empirical evidence for the efficacy of the proposed regularization term, we validate our algorithm using a controlled 2D toy experiment. As illustrated in Fig. 9-(a), the toy experiment utilizes Gaussian Mixture Models distributed uniformly in 2D space to represent the source data (green), target data (blue), and noise (red) distributions. Since direct source and target prompt conditioning is infeasible in this setting, separate Rectified Flow models were trained for the noise-to-source and noise-to-target data mappings. We compare the transport performance across three methodologies:

- Inversion and Sampling: Fig. 9-(b) and 9-(b') demonstrate that the sequential application of source-to-noise inversion followed by noise-to-target sampling successfully transports data from the source domain to the target domain along clear trajectories.

- FlowEdit: As FlowEdit samples random noise at every time step, this random noise selection frequently leads to out-of-domain outputs (Fig. 9-(c)) and results in non-straight transport trajectories (Fig. 9-(c')).

- FlowAlign: Figures 9-(d) illustrates the effect of incorporating our regularization term, which is derived from the first-order approximation of the optimal control solution. This regularization significantly reduces the prevalence of out-of-domain samples (Fig. 9-(d)) compared to FlowEdit. Crucially, FlowAlign yields straighter transport paths (Fig. 9-(d')) and effectively ensures that source data is transported to close target data, simplifying the complex trajectory in Fig. 9-(b').

The reliance on the first-order velocity approximation and the multi-step nature of the flow-based model introduces some failure cases. As depicted in Fig. 9-(d''), when the predicted velocity field

Table 3: Quantitative results for image editing on PIEBench, compared at identical ODE timesteps. (CFG scale = 13.5)

| method | Source Consistency | | | | | Semantic Alignment | |
|---|---|---|---|---|---|---|---|
| | Structural Distance ↓ | Background PSNR ↑ | Background LPIPS ↓ | Background MSE ↓ | Background SSIM ↑ | CLIP ↑ | HPS(x10²) ↑ |
| DDIB (Su et al., 2023) | 0.124 | 13.22 | 0.312 | 0.064 | 0.672 | 24.17 | 19.727 |
| SDEdit (Meng et al., 2021) | 0.037 | 22.17 | 0.127 | 0.008 | 0.740 | 24.84 | 20.837 |
| RF-Inv (Rout et al., 2025a) | 0.111 | 15.12 | 0.308 | 0.038 | 0.633 | 27.67 | 21.289 |
| FlowEdit (Kulikov et al., 2024) | 0.041 | 21.44 | 0.108 | 0.009 | 0.814 | 26.46 | 21.106 |
| FlowAlign | 0.031 | 24.21 | 0.069 | 0.005 | 0.864 | 25.65 | 20.850 |

Table 4: Quantitative results for image editing on PIEBench, compared at identical ODE timesteps. (CFG scale = 13.5)

| method | Source Consistency | | | | | Semantic Alignment | |
|---|---|---|---|---|---|---|---|
| | Structural Distance ↓ | (bg) PSNR ↑ | (bg) LPIPS ↓ | (bg) MSE ↓ | (bg) SSIM ↑ | CLIP ↑ | HPS(x10²) ↑ |
| DDIB | 0.124 | 13.22 | 0.312 | 0.064 | 0.672 | 24.17 | 19.727 |
| SDEdit | 0.037 | 22.17 | 0.127 | 0.008 | 0.740 | 24.84 | 20.837 |
| RF-Inv | 0.111 | 15.12 | 0.308 | 0.038 | 0.633 | **27.67** | **21.289** |
| FlowEdit | 0.041 | 21.44 | 0.108 | 0.009 | 0.814 | 26.46 | 21.106 |
| Ours | **0.031** | **24.21** | **0.069** | **0.005** | **0.864** | 25.65 | 20.850 |

points in an incorrect direction during the initial timesteps (low signal-to-noise ratio), the generated sample trajectory can diverge, leading to out-of-domain samples. Although this occurrence is observed less frequently in FlowAlign (Fig. 9-(d)) compared to FlowEdit (Fig. 9-(c)).

We provide an additional visualization in Fig. 21 comparing the velocity vector field (dx) and the resulting trajectory (xt) of FlowEdit and FlowAlign at the same timestep t. Without an explicit regularization term for source consistency, the velocities generated by FlowEdit appear noisy and dispersed. In sharp contrast, the velocities generated by FlowAlign are clean and concentrated near the specified editing target. Consequently, the accumulated change from the source image in FlowEdit's editing results is significant and widespread, whereas the results produced by FlowAlign remain close to the source while only applying the desired edit.

# F ADDITIONAL EVALUATION RESULTS ON PIEBENCH

## F.1 QUANTITATIVE AND QUALITATIVE RESULTS

We provide the complete evaluation result on PIEBench in Table 4, which includes metrics described in Section B. As discussed in the main paper, the proposed method outperforms all baselines in source consistency metrics while achieving both CLIP (ViT-L/14) and HPS scores comparable to FlowEdit. Although RF-Inversion, an inversion-based editing algorithm, exhibit higher semantic alignment scores, they tend to overemphasize the target concept at the expense of preserving the source structure (see more examples in Fig. 10 and 11). We also illustrate more qualitative results of the proposed method on text-based image editing task. Specifically, Fig. 10 and 11 present additional qualitative comparison between baselines and the FlowAlign. For diverse editing categories and objects, the proposed method shows better editing capability with better source structure preservation.

We additionally provide a comparison under a fixed computational budget in Table 5. All methods are benchmarked at approximately 99 Neural Function Evaluations (NFEs) to match the computational cost of our approach. For the inversion-based method, we used 33 ODE timesteps for inversion without CFG and 33 for sampling. For SDEdit and FlowEdit, which require a specified initial Signal-to-Noise Ratio (SNR), we selected starting timesteps that align with the noise levels proposed in FlowEdit (Kulikov et al., 2024), initiating them at timestep 26 (out of 76) and 13 (out of 37), respectively, totaling 100 NFEs. The results under this equalized budget are consistent with our previous findings regarding source consistency and HPS. However, this analysis also reveals that while baseline methods can achieve higher CLIP scores with more function evaluations, it leads to a significant degradation in source preservation. This reveals a fundamental inability of these approaches to maintain structural fidelity when striving for better semantic alignment, a limitation our method successfully overcomes.

Table 5: Quantitative results for image editing on PIEBench, compared using an identical Neural Function Evaluations (NFEs). (CFG scale = 13.5)

| method | Source Consistency | | | | | Semantic Alignment | |
|---|---|---|---|---|---|---|---|
| | Structural Distance ↓ | Background PSNR ↑ | Background LPIPS ↓ | Background MSE ↓ | Background SSIM ↑ | CLIP ↑ | HPS(x$10^2$) ↑ |
| DDIB (Su et al., 2023) | 0.137 | 12.03 | 0.368 | 0.083 | 0.609 | 26.69 | 20.459 |
| SDEdit (Meng et al., 2021) | 0.073 | 16.95 | 0.254 | 0.025 | 0.634 | 26.33 | 20.837 |
| RF-Inv (Rout et al., 2025a) | 0.121 | 13.10 | 0.361 | 0.060 | 0.572 | 27.26 | 20.837 |
| FlowEdit (Kulikov et al., 2024) | 0.041 | 21.23 | 0.113 | 0.100 | 0.808 | 26.46 | 21.057 |
| FlowAlign | 0.031 | 24.21 | 0.069 | 0.005 | 0.864 | 25.65 | 20.850 |

Table 6: CLIP score on all PIEBench Categories

| Category | DDIB | FlowEdit | RF-Inversion | SDEdit | Ours |
|---|---|---|---|---|---|
| 0-random | 26.72 | 25.48 | 24.77 | 26.08 | 27.41 |
| 1-change object | 23.95 | 26.44 | 27.33 | 24.81 | 25.30 |
| 2-add object | 25.46 | 27.18 | 28.25 | 25.87 | 26.42 |
| 3-delete object | 22.50 | 24.82 | 26.50 | 23.46 | 23.69 |
| 4-change attribute content | 23.44 | 26.28 | 26.63 | 24.32 | 25.20 |
| 5-change attribute pose | 23.66 | 26.21 | 26.96 | 25.24 | 25.75 |
| 6-change attribute color | 24.89 | 26.52 | 28.70 | 24.66 | 26.19 |
| 7-change attribute material | 24.13 | 26.39 | 27.81 | 24.88 | 25.98 |
| 8-change background | 24.03 | 25.98 | 27.16 | 24.45 | 25.47 |
| 9-change style | 25.30 | 28.14 | 29.59 | 26.83 | 27.22 |

## F.2 ABLATION

The proposed method includes two hyper-parameters: $\omega$, which controls efficient CFG, and $\zeta$, which weights the source consistency term derived from the flow-matching regularization. As discussed in Section 4 of the main paper, there is a trade-off between semantic alignment with the target text and structural consistency with the source image. Fig. 12 presents qualitative examples from the ablation study, focusing on the effect of $\zeta$ with a fixed $\omega = 10.0$. While the editing results may vary across samples even for the same $\zeta$, we find that $\zeta = 0.01$ consistently yields robust and balanced performance, effectively satisfying both the editing instruction and source structure preservation across diverse cases.

Additionally we compare robustness of the proposed hyperparameters in FlowAlign and FlowEdit in Fig. 13. In FlowAlign (Fig. 13-(a)), $\zeta = 0.01$ shows robust performance on the trade-off between CLIP similarity and structural distance across various values of $\omega$. In contrast, the original FlowEdit formulation (Fig. 13-(b)) lacks a clear Pareto optimal curve when evaluating the impact of its signal to noise ratio $N$. This suggests a non-robust relationship between $N$ and the editing outcomes. FlowEdit relies on a heuristically determined value (e.g., $N = 0.34$ in 50 timesteps) for early stopping, which is not systematically optimized for balancing these two objectives.

## F.3 FAILURE CASES

While FlowAlign achieves competitive performance across most PIEBench categories, it exhibits its lowest CLIP score in the "3-delete object" category (Tab. F.3). Detailed qualitative examples illustrating these limitations are provided in Fig. 22. The challenge of precise object deletion often results in FlowAlign generating an unwanted replacement in the original object's location. This issue is not unique to our approach; both FlowEdit and SDEdit exhibit similar failure modes. Furthermore, FlowEdit frequently compromises overall image fidelity by introducing unintended transformations to regions outside the specified editing area. In contrast, while RF-Inversion and DDIB more closely adhere to the textual editing instructions for deletion, they fail to maintain satisfactory source image consistency, leading to significant content deviation from the original image.

Table 7: Quantitative Comparison on video editing

|  | FlowEdit | FlowAlign | FlowDirector (FE+mask) | FlowAlign + mask |
|---|---|---|---|---|
| Background PSNR | 26.73 | **27.19** | 25.39 | 26.46 |
| Background LPIPS | 0.0719 | **0.0662** | 0.0765 | 0.0702 |
| Background MSE | 0.0032 | **0.0028** | 0.0043 | 0.0037 |
| Background SSIM | 0.8167 | 0.8286 | 0.8197 | **0.8340** |
| Overall Consistency | 0.2164 | 0.2209 | **0.2386** | 0.2276 |
| CLIP | 24.18 | 24.22 | **25.41** | 24.85 |

# G  ADDITIONAL RESULTS ON VIDEO AND 3D EDITING

As we demonstrated in the last section of the main paper, FlowAlign can be extended beyond text-based image editing. In this section, we provide details and additional examples for video editing and 3D editing via Gaussian splatting.

## G.1  FURTHER APPLICATIONS - VIDEO EDITING

FlowAlign is fundamentally a text-based image editing method built on an image flow-based model. In contrast, video editing typically relies on generative models trained on video datasets with temporal attention mechanisms to enhance temporal consistency (Jeong et al., 2024b; Liu et al., 2024; Jeong et al., 2024a; Park et al., 2025). However, video editing can also be viewed as a sequence of image editing tasks. Extending the applicable range of image editing methods to video could offer practical benefits, such as reduced training and inference costs.

While the proposed method does not explicitly enforce temporal consistency, we can still apply it independently to each video frame. For this experiment, we use the DAVIS dataset (Caelles et al., 2019) and extract the source text prompt using LLaVA (Liu et al., 2023a), conditioned on the middle frame of each video. For the hyperparameter $\lambda$, we use a constant of 0.01, consistent with the main experiments. For $\omega$, we vary it over [5.0, 7.5, 10.0, 13.5] and qualitatively select the best-performing value.

Figures 14 and 15 present examples of edited video frames, focusing on texture and object editing tasks. Due to the strong source structure consistency of the proposed method, the video background remains well-preserved. However, temporal consistency for the edited object is limited, as no explicit constraint is imposed- for example, the head of the swan in Figure 14. Nonetheless, these results highlight the potential of the proposed method as a lightweight solution for video editing.

### G.1.1  QUANTITATIVE COMPARISON ON VIDEO EDITING

We evaluate FlowAlign against FlowEdit and FlowDirector Li et al. (2025) on 125 cases (25 videos with 5 different editing prompts). All methods were tested on the DAVIS dataset Pont-Tuset et al. (2017) utilizing the latent video model, WAN2.1-T2V Wan et al. (2025). We also include results for FlowAlign combined with attention masking introduced in FlowDirector. Tab. 7 demonstrates that FlowAlign achieves superior source consistency compared to baselines while maintaining comparable text alignment results.

### G.1.2  QUALITATIVE COMPARISON ON VIDEO EDITING

In Fig. 16, FlowAlign demonstrates improved text-to-video alignment compared to FlowEdit, which relies on an early-stopping mechanism. Furthermore, FlowAlign exhibits superior source consistency when compared to FlowDirector (e.g., observe the preservation of the subject's face in the frame on the right).

## G.2  FURTHER APPLICATIONS - GAUSSIAN SPLATTING EDITING

Recall that one of the main contributions of FlowAlign is introducing a regularization term for inversion-free image editing method, which has a goal of simulating the flow ODE between two

Table 8: Quantitative Comparison on 3DGS editing

|                  | SDS  | DDS  | PDS  | IGS2GS | FlowAlign |
|------------------|------|------|------|--------|-----------|
| (bg) LPIPS       | 0.33 | 0.33 | 0.23 | 0.10   | **0.06**  |
| CLIP Dir. Sim.   | 0.10 | 0.18 | 0.12 | 0.07   | **0.18**  |
| CLIP             | **0.27** | 0.27 | 0.24 | 0.23 | 0.26    |
| Aesthetic Score  | 4.94 | 4.76 | 4.12 | 5.48   | **5.52**  |

image samples. Accordingly, we can apply FlowAlign for a 3D editing via Gaussian splatting. The basic idea is to distill the knowledge of text-conditioned image prior of flow model to update Gaussians.

The common way to distill the prior knowledge to parameters of generators, such as NeRF or Gaussian Splatting, is to optimize those parameters with score distillation gradients. More generally, it involves guiding the parameters through the use of a pre-trained denoiser acting as a critic. From the result of Proposition 1, we construct a regularized trajectory between source and target images and the drift term Eq. (45) corresponds to gradient reflecting the editing direction. Specifically, we can define a editing loss function that satisfies

$$\nabla_{\boldsymbol{x}_t}\mathcal{L}_{FA} := \boldsymbol{v}_t(\boldsymbol{p}_t, \boldsymbol{c}_{tgt}) - \boldsymbol{v}_t(\boldsymbol{q}_t, \boldsymbol{c}_{src}) + \gamma(\mathbb{E}[\boldsymbol{p}_0|\boldsymbol{p}_t] - \mathbb{E}[\boldsymbol{q}_0|\boldsymbol{q}_t]), \quad (46)$$

where we assume that the Jacobians $\frac{\partial \boldsymbol{v}_t(\boldsymbol{q}_t)}{\partial \boldsymbol{x}_t}$ and $\frac{\partial \boldsymbol{v}_t(\boldsymbol{p}_t)}{\partial \boldsymbol{x}_t}$ are identity matrices. Then, by using $\mathcal{L}_{FA}$ as a critic for updating Gaussians, we can guide them toward target Gaussian whose rendered views are high likely sample in perspective of flow model. The only one we should consider is extending $\mathcal{L}_{FA}$ to contain parameters $\psi$ of differentiable generator $\boldsymbol{g}$.

Because we use flow model defined in latent space, we compute the loss function using $\boldsymbol{q}_{t,c} = (1-t)\boldsymbol{x}_{src,c} + t\boldsymbol{\epsilon}$, $\boldsymbol{p}_{t,c} = \boldsymbol{q}_{t,c} + \boldsymbol{x}_{t,c} - \boldsymbol{x}_{src,c}$. Here, $c$ denotes sampled camera view, $\boldsymbol{x}_{src,c} = \mathcal{E}(\boldsymbol{g}(\psi_{src}, c))$ denotes the latent code of rendered view from initial Gaussian, $\boldsymbol{x}_{t,c} = \mathcal{E}(\boldsymbol{g}(\psi, c))$ denotes the latent code of rendered vidw from current Gaussian, $\mathcal{E}$ denotes the pre-trained encoder of a VAE and $\boldsymbol{\epsilon} \sim \mathcal{N}(0, \boldsymbol{I})$. By using the chain rule, we obtain the following gradient for flow-based 3D editing:

$$\nabla_{\psi}\mathcal{L}_{FA} := [\nabla_{\boldsymbol{x}_t}\mathcal{L}_{FA}] \frac{\partial \boldsymbol{x}_t}{\partial \psi} \quad (47)$$

$$= [\boldsymbol{v}_t(\boldsymbol{p}_{t,c}, \boldsymbol{c}_{tgt}) - \boldsymbol{v}_t(\boldsymbol{q}_{t,c}, \boldsymbol{c}_{src}) + \gamma(\mathbb{E}[\boldsymbol{p}_0|\boldsymbol{p}_t] - \mathbb{E}[\boldsymbol{q}_0|\boldsymbol{q}_t])] \frac{\partial \boldsymbol{x}_t}{\partial \psi}, \quad (48)$$

and use gradient descent to update the $\psi$ as

$$\psi = \psi - \eta_t \nabla_{\psi}\mathcal{L}_{FA}. \quad (49)$$

For the semantic guidance term in this gradient, we only apply CFG for $\boldsymbol{p}_t$ by setting $c_{src}$ as null-text embedding, as same as the image editing algorithm. The pseudocode for the 3D editing with FlowAlign is in Algorithm 2, and additional qualitative results are in Fig. 18.

### G.2.1 QUANTITATIVE COMPARISON ON 3D EDITING

We perform Gaussian Splatting optimization for $3,000$ iterations using a classifier-free guidance (CFG) weight $\omega$ set to either 70 or 100. Experiments are conducted on real-world scenes using datasets from IN2N (Haque et al., 2023) and PDS (Koo et al., 2024). For comparison, we benchmark against existing distillation methods, including SDS (Poole et al., 2022), DDS (Hertz et al., 2023), and PDS (Koo et al., 2024), as well as the non-distillation baseline IGS2GS (Vachha & Haque, 2024). For baselines using distillation methods we used Stable Diffusion 3.0 and for IGS2GS (Vachha & Haque, 2024), we use UltraEdit (Zhao et al., 2024) in replacement of Instruct-Nerf2Nerf (Haque et al., 2023) for flow-based models. Across 15 scenes (6 distinct scenes with varying prompts), FlowAlign demonstrates superior performance not only in source consistency (measured by background LPIPS) but also in 3D consistency (measured by CLIP directional consistency) and human preference (measured by LAION aesthetic score) (Tab. 8).

---

**Algorithm 2** Algorithm of FlowAlign for 3D editing

---

**Require:** Source parameter $\psi_{src}$, Pre-trained flow-based model $v_\theta$, VAE encoder $\mathcal{E}$, Source/Target text embeddings $\boldsymbol{c}_{src}, \boldsymbol{c}_{tgt}$, CFG scale $\omega$, camera views $Cams$, Noise Schedule $\sigma_t$, differentiable generator $\boldsymbol{g}$, min and max timestep $T_{min}, T_{max}$

1: $\psi \leftarrow \psi_{src}$
2: *// Freeze VAE encoder $\mathcal{E}$ and flow model $v_\theta$*
3: $\boldsymbol{x}_{src,c} \leftarrow \mathcal{E}(\boldsymbol{g}(\psi_{src}, c))$
4: **for** $t : T_{max} \rightarrow T_{min}$ **do**
5: $\quad \epsilon \sim \mathcal{N}(0, \sigma^2 \mathbf{I}) \quad c \sim \mathcal{U}(Cams)$
6: $\quad \boldsymbol{x}_{t,c} \leftarrow \mathcal{E}(\boldsymbol{g}(\psi, c))$
7: $\quad \boldsymbol{q}_{t,c} \leftarrow (1 - \sigma_t)\boldsymbol{x}_{src,c} + \sigma_t \epsilon$
8: $\quad \boldsymbol{p}_{t,c} \leftarrow \boldsymbol{x}_{t,c} - \boldsymbol{x}_{src,c} + \boldsymbol{q}_{t,c}$
9: $\quad \boldsymbol{v}^\theta(\boldsymbol{p}_{t,c}) = \boldsymbol{v}^\theta(\boldsymbol{p}_{t,c}, \boldsymbol{c}_{src}) + \omega \left[ \boldsymbol{v}^\theta(\boldsymbol{p}_{t,c}, \boldsymbol{c}_{tgt}) - \boldsymbol{v}^\theta(\boldsymbol{p}_{t,c}, \boldsymbol{c}_{src}) \right], \quad \boldsymbol{v}^\theta(\boldsymbol{q}_{t,c}) = \boldsymbol{v}^\theta(\boldsymbol{q}_{t,c}, \boldsymbol{c}_{src})$
10: $\quad \mathbb{E}[\boldsymbol{p}_0 | \boldsymbol{p}_t] \leftarrow \boldsymbol{p}_{t,c} - t\boldsymbol{v}^\theta(\boldsymbol{p}_{t,c}), \quad \mathbb{E}[\boldsymbol{q}_0 | \boldsymbol{q}_t] \leftarrow \boldsymbol{q}_{t,c} - t\boldsymbol{v}^\theta(\boldsymbol{q}_{t,c})$
11: $\quad \nabla_\psi \mathcal{L}_{FA} \leftarrow [(\boldsymbol{v}_\theta(\boldsymbol{p}_{t,c}) - \boldsymbol{v}_\theta(\boldsymbol{q}_{t,c}) + \gamma(\mathbb{E}[\boldsymbol{p}_0 | \boldsymbol{p}_t] - \mathbb{E}[\boldsymbol{q}_0 | \boldsymbol{q}_t])\frac{\partial \boldsymbol{x}}{\partial \psi}]$
12: $\quad \psi \leftarrow \psi - \eta_t \nabla_\psi \mathcal{L}_{FA}$
13: **end for**
14: **return** $\psi$

---

**Algorithm 3** Algorithm of FlowAlign with FLUX (Guidance distilled model)

---

**Require:** Source image $\boldsymbol{x}_{src}$, Pre-trained flow model $\boldsymbol{v}^\theta$, VAE encoder and Decoder $\mathcal{E}, \mathcal{D}$, Source/Target text embeddings $c_{src}, c_{tgt}$, CFG scales $\omega_{src}, \omega_{tgt}$, source consistency scale $\zeta$

1: $\boldsymbol{x}_t \leftarrow \mathcal{E}(\boldsymbol{z}_{src})$
2: **for** $t : 1 \rightarrow 0$ **do**
3: $\quad \epsilon \sim \mathcal{N}(0, \mathbf{I})$
4: $\quad \boldsymbol{q}_t \leftarrow (1 - t)\boldsymbol{x}_{src} + t\boldsymbol{\epsilon}$
5: $\quad \boldsymbol{p}_t \leftarrow \boldsymbol{x}_t - \boldsymbol{x}_{src} + \boldsymbol{q}_t$
6: $\quad \boldsymbol{v}^\theta(\boldsymbol{p}_t) := \boldsymbol{v}^\theta(\boldsymbol{p}_t, \boldsymbol{c}_{tgt}, \omega_{tgt}), \quad \boldsymbol{v}^\theta(\boldsymbol{q}_t) := \boldsymbol{v}^\theta(\boldsymbol{q}_t, \boldsymbol{c}_{src}, \omega_{src})$
7: $\quad \mathbb{E}[\boldsymbol{p}_0 | \boldsymbol{p}_t] \leftarrow \boldsymbol{p}_t - t\boldsymbol{v}^\theta(\boldsymbol{p}_t), \quad \mathbb{E}[\boldsymbol{q}_0 | \boldsymbol{q}_t] \leftarrow \boldsymbol{q}_t - t\boldsymbol{v}^\theta(\boldsymbol{q}_t)$
8: $\quad \boldsymbol{x}_t \leftarrow \boldsymbol{x}_t + \left[ \boldsymbol{v}^\theta(\boldsymbol{p}_t) - \boldsymbol{v}^\theta(\boldsymbol{q}_t) \right] dt + \zeta(\mathbb{E}[\boldsymbol{q}_0 | \boldsymbol{q}_t] - \mathbb{E}[\boldsymbol{p}_0 | \boldsymbol{p}_t])$
9: **end for**
10: $\boldsymbol{z}_{edit} \leftarrow \mathcal{D}(\boldsymbol{x}_t)$

---

### G.2.2 QUALITATIVE COMPARISON ON 3D EDITING

Qualitative comparison of our method against score distillation methods (SDS, DDS, PDS) and IGS2GS are provided in Fig. 17.

## H FURTHER VARIANTS - FLUX

Our main experiments are conducted using by leveraging pre-trained flow model, Stable Diffusion 3.0. Since the proposed method regulates the sampling ODE via a flow matching cost, it is also compatible with other flow-based models, such as FLUX. Following the setup in FlowEdit (Kulikov et al., 2024), we additionally evaluate our method using FLUX as the backbone model. We incorporate the source consistency term derived from optimal control with flow matching regularization, as described in Algorithm 3. As FLUX.1-dev is a guidance-distilled model that directly takes the CFG scale as input, there is no need to apply the modified CFG strategy used in Stable Diffusion 3.0 case. Therefore, we simply provide the CFG scale to both $\boldsymbol{v}(\boldsymbol{p}_t)$ and $\boldsymbol{v}(\boldsymbol{q}_t)$, following the approach used in FlowEdit. Figure 19 illustrates edited results generated by the proposed method implemented with FLUX.1-dev. Similar to the results with Stable Diffusion 3.0, the outputs effectively reflect the intended editing direction specified by the text prompts while preserving source structures. These results imply that the proposed method is broadly applicable to flow-based models for image editing.

Table 9: Quantitative Comparison on additional image editing baselines on FLUX

|  | FireFlow | FluxSpace | RF-Solver-Edit | FlowAlign |
|---|---|---|---|---|
| Structure | 0.0456 | 0.1647 | 0.0451 | **0.0414** |
| Background PSNR | 20.37 | 11.69 | 20.34 | **20.75** |
| Background LPIPS | 0.1824 | 0.3914 | 0.1824 | **0.1314** |
| Background MSE | 0.0136 | 0.0971 | 0.0136 | **0.0122** |
| Background SSIM | 0.7584 | 0.5747 | 0.7589 | **0.8123** |
| CLIP | 26.69 | 24.41 | 26.75 | **26.97** |

## H.1 QUANTITATIVE COMPARISON ON FLUX

We provide additional comparison against methods that rely on source image inversion - RF-Edit Wang et al. (2025), FluxSpace Dalva et al. (2024) and FireFlow Deng et al. (2024). Our results show that FlowAlign achieves more faithful and controlled edits than the inversion-based baselines. Inversion-based methods inherently suffer from reconstruction errors, which lead to significant deviations from the source image. To mitigate this, RF-Edit, QK-Edit, and FireFlow inject attention-layer features (Q/K/V) extracted during the inversion process back into the generation process. FluxSpace leverages an editing-direction attention shift. However, as shown in Tab. 9, all remain highly sensitive to inversion errors; even with higher-order ODE solvers, flow inversion is imperfect and computationally expensive. FlowEdit and FlowAlign bypasses inversion altogether, avoiding these errors without adding computational overhead, while FlowAlign further enhances source consistency with optimal control formulation and source consistency regularization.

## H.2 QUALITATIVE COMPARISON ON FLUX

In Fig. 20, we illustrate qualitative comparison on Flux. While additional baselines - FireFlow Deng et al. (2024), FluxSpace Dalva et al. (2024) and RF-Solver-Edit Wang et al. (2025) result in saturated images, FlowAlign shows superior source preservation with better editing capability.

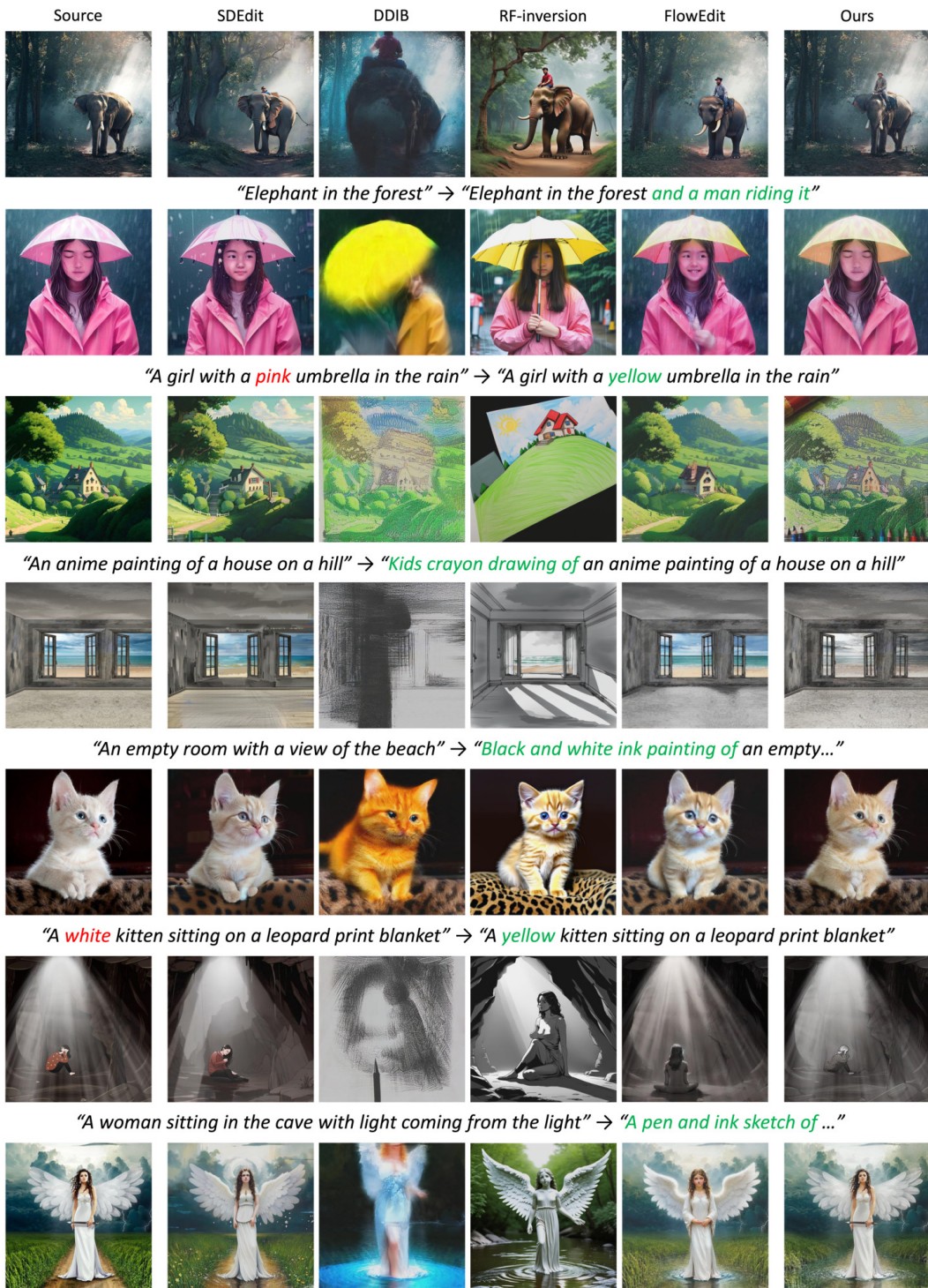

Figure 10: Additional qualitative comparison results.

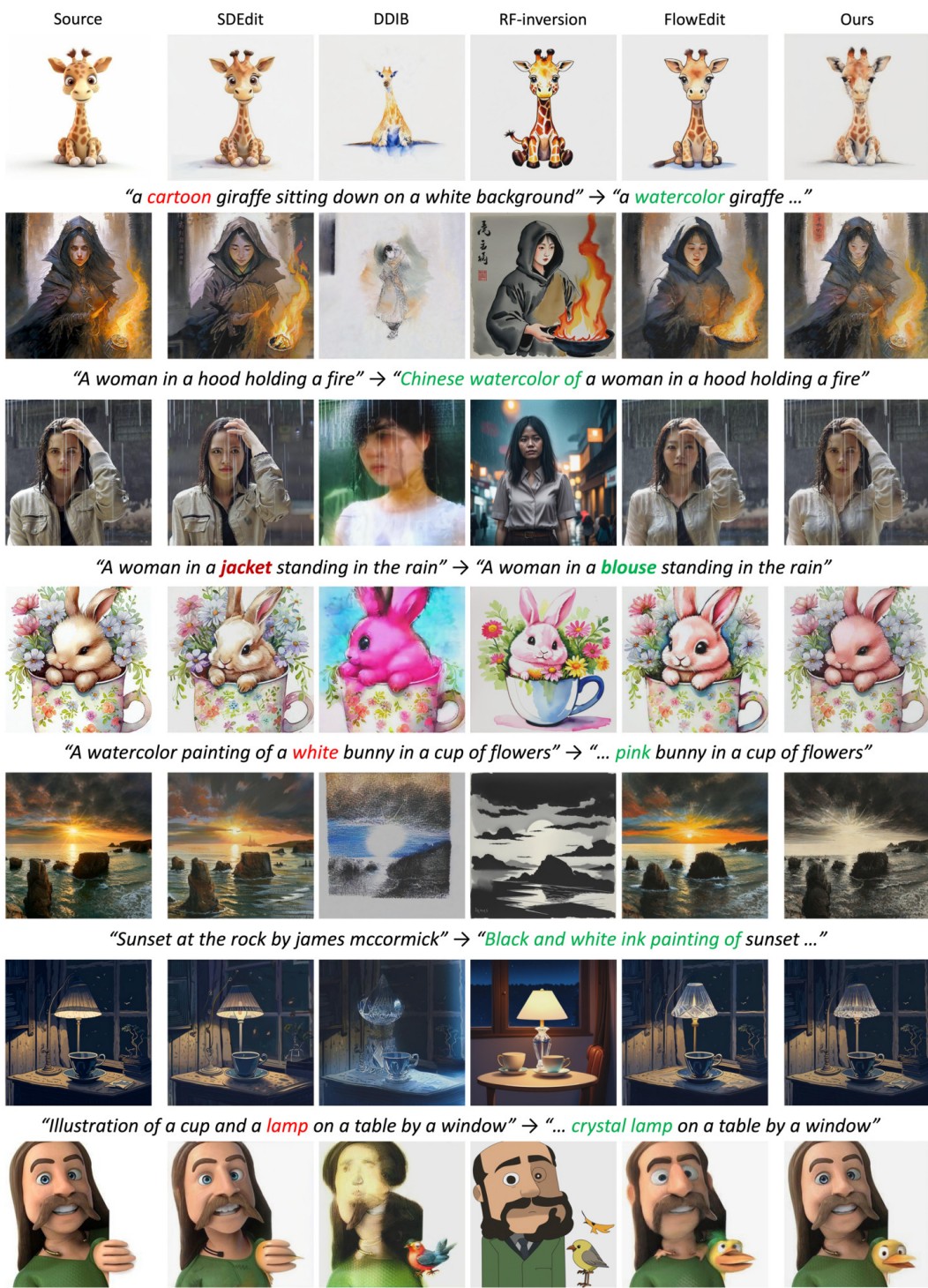

Figure 11: Additional qualitative comparison results.

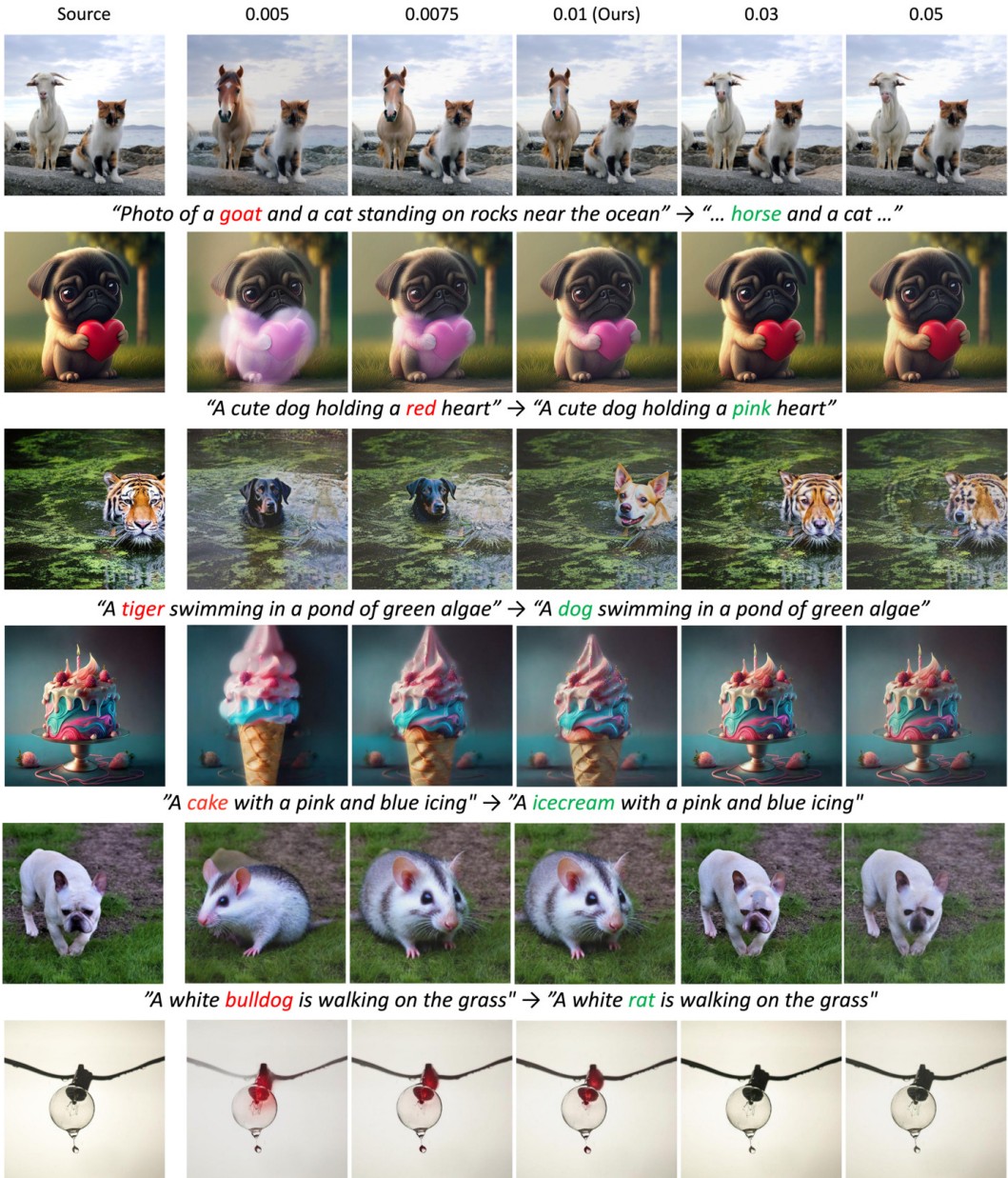

Figure 12: Ablation results for $\zeta$.

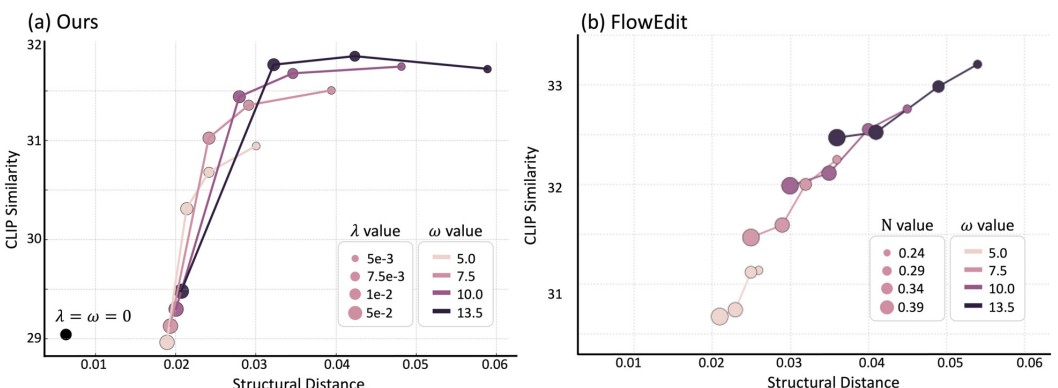

Figure 13: Ablation study for hyperparameters of FlowAlign and FlowEdit.

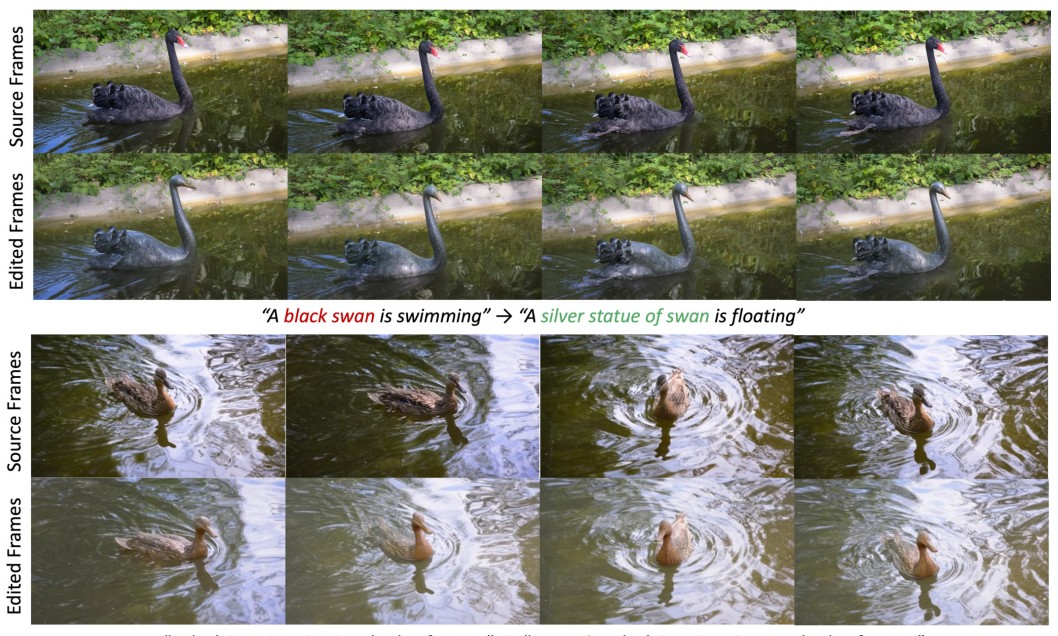

Figure 14: Additional results for video editing: texture change.

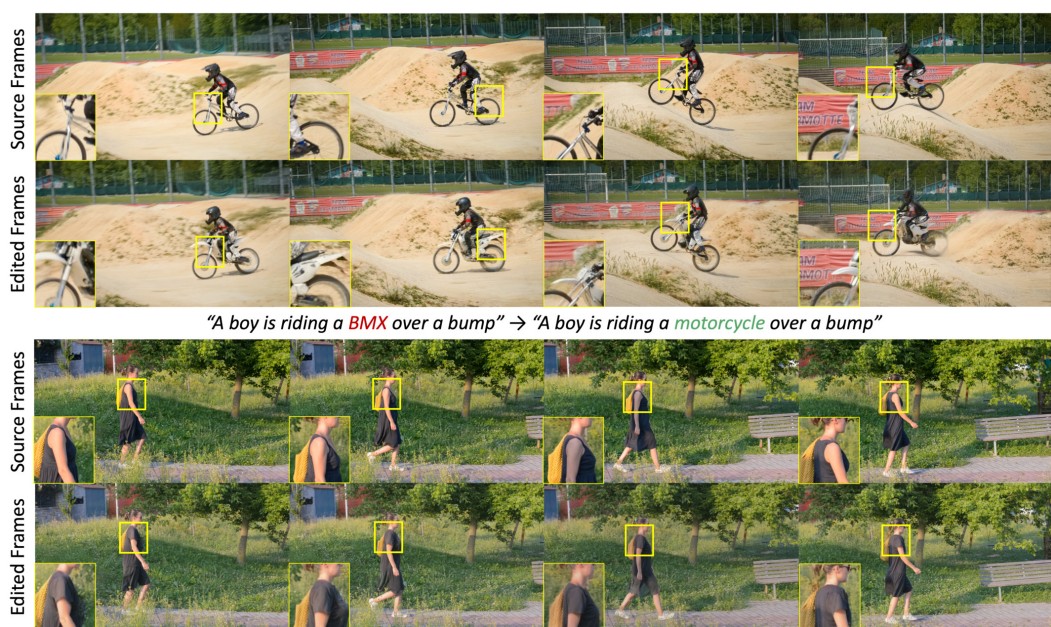

Figure 15: Additional results for video editing: object change.

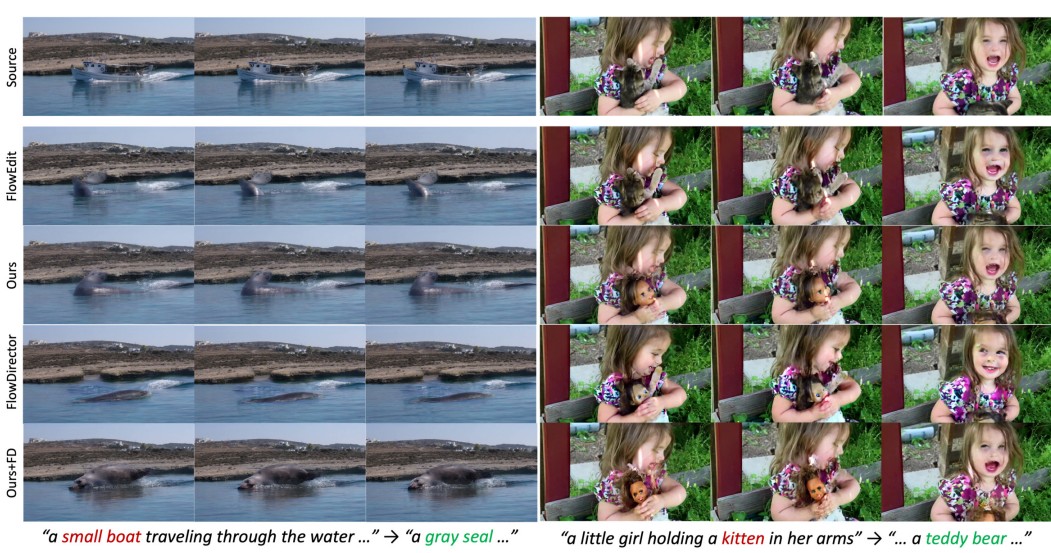

Figure 16: Qualitative Comparison on video editing.

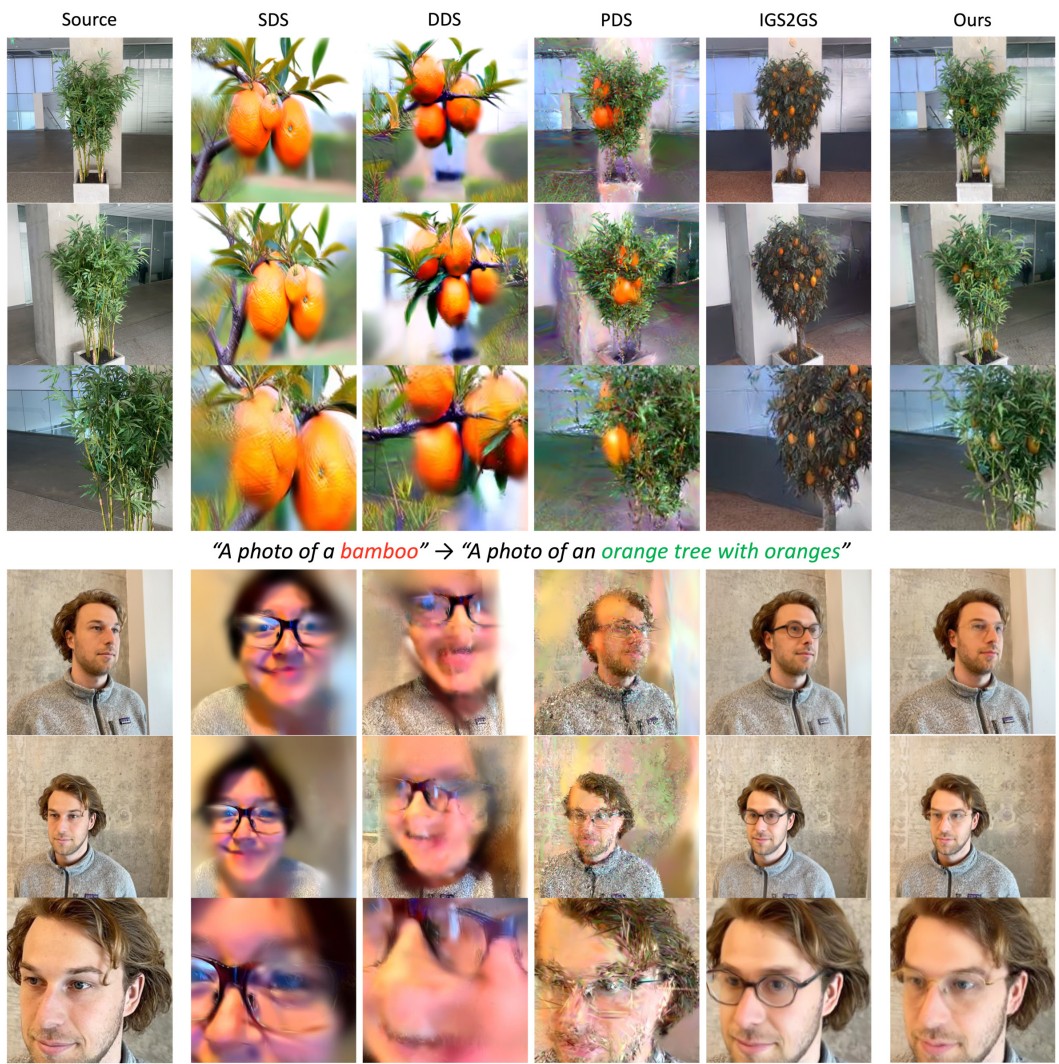

Figure 17: Additional results for 3D editing.

Figure 18: Video results of 3D editing. *Click each image to play the video in Acrobat Reader.*

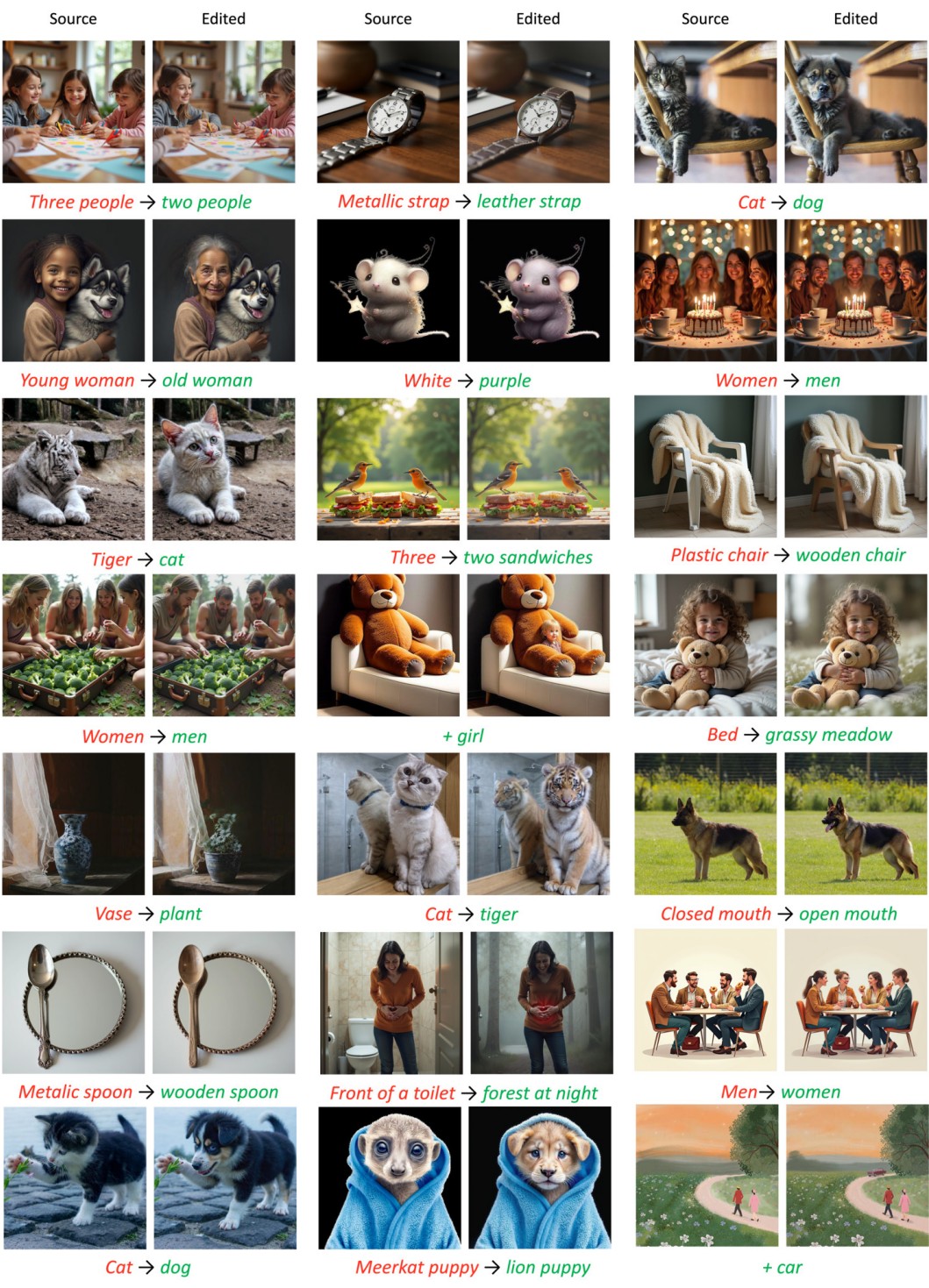

Figure 19: Qualitative image editing results using FLUX.

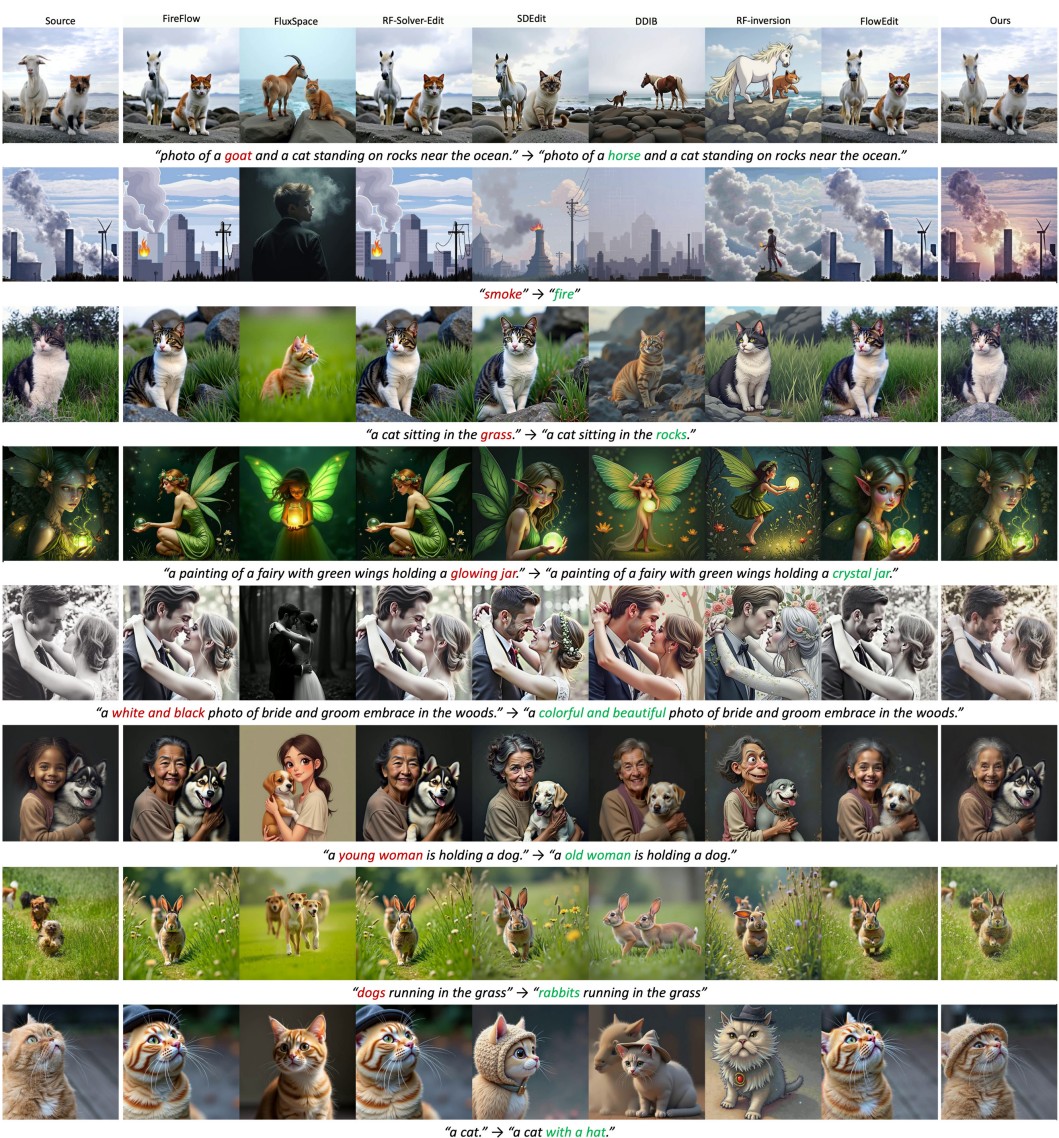

Figure 20: Qualitative Comparison with baseline methods on FLUX.

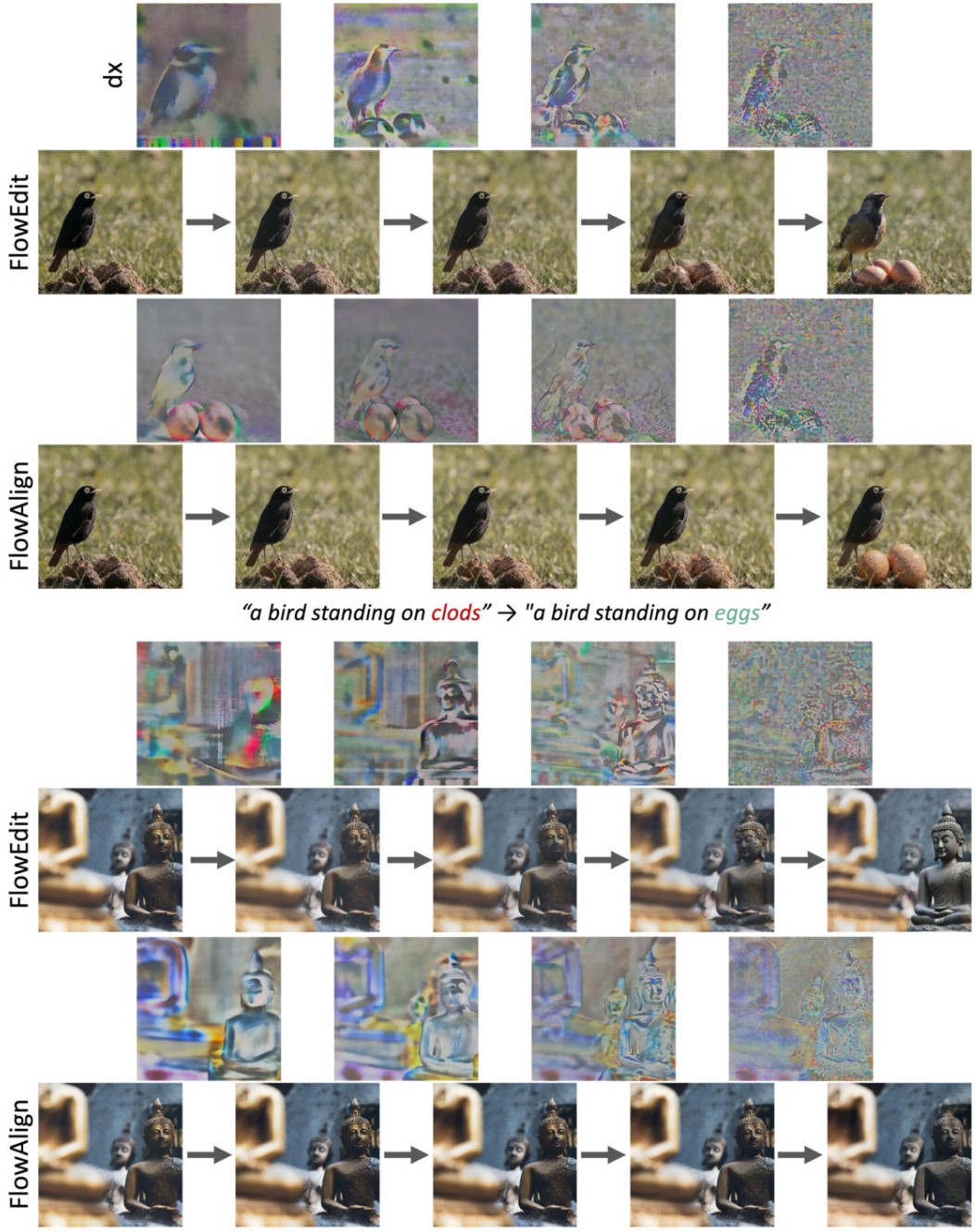

Figure 21: Visualization of FlowEdit and FlowAlign editing trajectories.

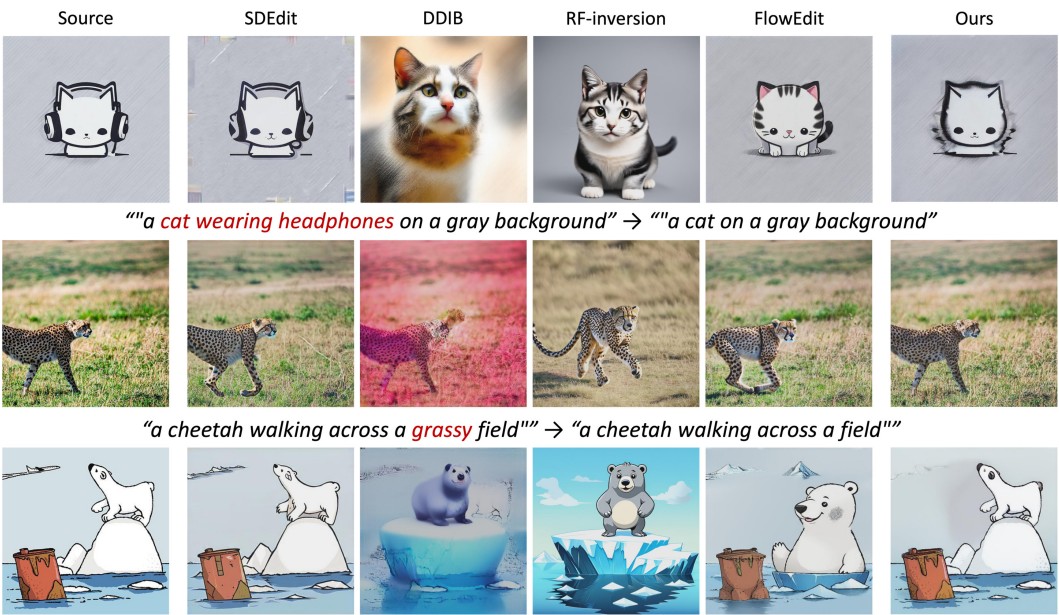

Figure 22: Failure cases of FlowAlign on "3-delete object" category in PIEBench.

