# OpenReview forum: "FlowAlign: Trajectory-Regularized, Inversion-Free Flow-based Image Editing"
_ICLR.cc/2026/Conference — ICLR 2026 Poster_

### Official Review · Reviewer_JBzF · 2025-10-27

**Soundness:** 2
**Presentation:** 2
**Contribution:** 2
**Rating:** 2
**Confidence:** 4

**Summary:**

FlowAlign is a flow-based approach for text-guided image editing that operates without additional training or inversion. Unlike prior methods such as FlowEdit, which suffer from unstable trajectories and source degradation, FlowAlign introduces trajectory regularization using terminal structural similarity to address these issues. This allows the method to achieve a balanced trade-off between semantic alignment and structural consistency, resulting in smooth and deterministic transformations. Moreover, FlowAlign achieves computational efficiency by requiring only one additional function evaluation (NFE) per ODE step and enables near-perfect reconstruction of the source image via backward ODE integration.

**Strengths:**

1.FlowAlign performs efficient editing without requiring inversion or additional training. In addition, FlowAlign is more efficient than FlowEditing.

2.FlowAlign achieves smooth and stable trajectories through trajectory regularization.

3.FlowAlign It balances semantic alignment and structural consistency effectively.

**Weaknesses:**

1.My main concern is the comparison with FlowCycle [1]. FlowCycle presents a more advanced approach than the proposed FlowAlign.

2.There are some questionable aspects in the figures and tables presented by the authors. Please refer to the "Questions" Section for specific details.

3.The diversity and explanation of CFG settings appear insufficient. Although the authors acknowledge that CFG is a highly critical parameter, they provide limited experimental results, using only fixed CFG settings for the baselines.

[1]FLOWCYCLE: PURSUING CYCLE-CONSISTENT FLOWS FOR TEXT-BASED EDITING

**Questions:**

1.The most important point is the comparison with FlowCycle. FlowCycle demonstrates superior performance compared to FlowAlign — can the authors explain why FlowAlign performs better than FlowCycle?

2.As mentioned in the Weaknesses Section, there are several questionable aspects in the presented results. The main ones are as follows:

2-1. In Figure 3(a), did the authors indeed set the CFG scales to {5.0, 7.5, 10.0, 13.5}? If so, were the source CFG and target CFG values identical for the baselines? If they were the same, this does not match the baseline settings described elsewhere (in other tables, figures, or the Implementation Details section). In particular, for FlowEdit, if the source CFG values were set to {5.0, 7.5, 10.0, 13.5}, as far as I know, this is not a standard setting.

2-2. While it is reasonable to point out the limitations of CLIP scores, the Human Evaluation results in Figure 3(b) and the CLIP scores in Table 3 are highly inconsistent. The discrepancy is too large to attribute merely to the limitations of CLIP score. Why does FlowAlign show better Human Evaluation results but lower CLIP scores?

3.The key idea of FlowAlign seems to be its improved efficiency by computing CFG only for 'v'. However, to convincingly demonstrate superior performance over the baselines, more comparisons involving 'source CFG' and 'target CFG' variations should be provided. How does the performance vary across baselines depending on the source and target CFG values?

---

> ### Author Response · Authors · 2025-11-24
> **Response by Authors**
>
> **W1 & Q1. Comparison with FlowCycle.**
>
> We kindly remind the reviewer that FlowCycle is follow-up work of FlowAlign, which opened even after the submission deadline of ICLR. Per the reviewer guidelines, requesting comparison or discussion with concurrent or subsequent works is not a valid basis for evaluation or rejection.
>
> However, we briefly outline the key distinctions, focusing on the efficiency and architectural differences between the two methods.
> - 1.	Optimization Paradigm: FlowAlign operates as an optimization-free method during inference, whereas FlowCycle is an optimization-based approach that requires iterative gradient updates at test time.
> - 2.	Computational Efficiency: FlowCycle incurs a significantly higher runtime cost. Its optimization phase requires an extensive computational budget, involving steps proportional to $\text{Epochs} \times \text{NFE} \times (\text{Source/Target Velocities}) \times (\text{Source/Target CFG Scales})$ with gradient back-propagation, and further inference cost. In contrast, FlowAlign achieves its result efficiently, requiring only $3 \times \text{NFE}$ during inference without any back-propagation.
> - 3.	Hyperparameter Complexity: FlowCycle introduces numerous additional hyperparameters, including separate CFG scales for training the source and target noise, a loss scale $\alpha$, an early-stopping time step similar to FlowEdit, and standard optimizer hyperparameters. Furthermore, FlowCycle tightly couples the CFG scale used during optimization with the scale used during inference, resulting in a less flexible framework compared to FlowAlign.
>
>
>
> **W2 and Q2. Clarification on Figures and tables.**
>
> Regarding Figure 3(a), we would like to clarify that we set the source CFG scale for FlowEdit to 3.5 by following the original paper, as mentioned in line 779 in the original manuscript.
>
> For the human preference evaluation, as in section C in appendix of the original manuscript, we have randomly sampled baseline methods and images. We have stated the limitation of CLIP score in lines 321-352 in the original manuscript such that CLIP score often fails to capture editing outcomes that are totally different from the source image. This inconsistency emphasizes the limitation of CLIP score as our claim instead of just a discrepancy. For example, see Figure 9 and 10 in the original manuscript, where RF-Inversion ranks the highest CLIP score of all methods, generating inconsistent results from the source images.
>
> **W3. Insufficient diversity and explanation for CFG setting.**
>
> We would like to clarify that we have used **four CFG scales** as the prior work FlowEdit did experiments with **three CFG scales**, which is not insufficient to show the trends of the editing performance. Importantly, our CFG range includes the recommended CFG scale by FlowEdit, which is not arbitrarily decided. Furthermore, we emphasize that we have stated details on CFG setting in lines 314-321 and 756-782 in the original manuscript, including the details on the source CFG scale of FlowEdit.
>
>
> **Q3. Ablation study on baseline performance with various CFG scales.**
>
> We would like to clarify that the main contribution of FlowAlign lies in formulating the optimal control problem with source-consistency regularization, rather than in efficiency with the CFG. Also, the CFG scale was varied for an ablation study to illustrate performance trends, not to claim optimal results across all possible CFG settings. Under the controlled experimental setup used in prior work (FlowEdit), we assure the reviewer that the presented experiments sufficiently demonstrate that FlowAlign improves source consistency while maintaining strong editing capability.

---

### Official Review · Reviewer_TFck · 2025-10-29

**Soundness:** 3
**Presentation:** 3
**Contribution:** 3
**Rating:** 6
**Confidence:** 4

**Summary:**

This paper introduces FlowAlign, a flow-based image editing framework that builds upon a prior inversion-free approach, FlowEdit. While this method demonstrated that direct editing without inversion is feasible, FlowAlign focuses on improving the smoothness and consistency of the editing trajectory. To achieve this, it introduces source similarity at the terminal point as a regularization term, encouraging trajectories that preserve structural consistency with the source image while following the intended semantic edits.

**Strengths:**

1. Good results with clear qualitative comparisons – the paper presents convincing visual examples that highlight the differences between FlowAlign and baselines. The figures clearly show improved source structure preservation and accurate semantic edits.
2. Human preference study – beyond quantitative metrics, the authors conduct a user study showing that participants consistently prefer FlowAlign’s outputs over baselines. As far as I know, this is the first time human preference is reported for these low-based editing methods.
3. Demonstrated across multiple settings – The method is evaluated on 2D image editing, video editing, and 3D Gaussian splatting, showing versatility and robustness across different tasks.

**Weaknesses:**

1. Trajectory regularization intuition – while trajectory regularization appears effective, the paper could provide a deeper explanation of why smoother trajectories improve editing outcomes. It seems intuitive that smoother trajectories help preserve the original image structure, but it is unclear what trade-offs or costs this introduces. For example, could this regularization limit the strength or flexibility of edits, or bias the model toward minimal changes?

2. Figure 2 clarity – figure 2 is somewhat confusing. It is not clear which panels correspond to FlowEdit versus FlowAlign. What does “R” represent? This term does not appear in the text. Additionally, the velocity field mentioned in the caption is not visually represented in the figure, making it difficult to interpret the intended trajectory differences. Clarifying these points would improve readability and understanding of the proposed method.

3. It is unclear whether the paper uses the same early-timestep skipping as FlowEdit.

**Questions:**

I raised question in the weaknesses section.

---

> ### Author Response · Authors · 2025-11-24
> **Response by Authors**
>
> **W1. Why do smoother editing trajectories improve editing outcomes?**
>
> We thank the reviewer for raising this important point. A smoother editing trajectory improves outcomes in two key aspects.
>
> First, it prevents excessive deviation from the original generative path - an issue introduced by DDPM-inversion-style noise injection [1]. Although this noise enables flexible image editing, it often causes overly saturated results unless parameters such as classifier-free guidance scale or early stopping are carefully tuned, as in FlowEdit. FlowAlign mitigates these issues naturally, without relying on such heuristics.
>
> Second, a smoother trajectory strengthens source consistency. While FlowEdit adheres well to editing instructions, it often loses key source attributes (e.g. identity or background) due to the absence of an explicit consistency mechanism. FlowAlign introduces a simple, theoretically grounded regularization term that preserves source information while maintaining strong editing capability.
>
>
> **W2. Clarity of Figure 2.**
>
> We appreciate pointing out unclear notation in Figure 2 in the original manuscript. R denotes the regularization term in Eq (22), which is an outcome of the proposed optimal control problem with source consistency. We fixed the caption of the Figure2 and added a definition of R in Eq (22).
>
> By considering source consistency term in the optimal control problem, we avoid excessive deviation from generative trajectory between the noise and source image, which results in smoother editing trajectory compared with FlowEdit.
>
>
> **W3. Clarification on the early-timestep skipping**
>
> We would like to clarify that FlowAlign **does not use** early-timestep skipping. Instead, we set the same number of timesteps with FlowEdit to compare the editing performance under the controlled experimental setting. Thus, the experimental results emphasize the effectiveness of source consistency in editing tasks.

---

### Official Review · Reviewer_GZwn · 2025-10-30

**Soundness:** 2
**Presentation:** 2
**Contribution:** 2
**Rating:** 2
**Confidence:** 4

**Summary:**

The paper proposes FlowAlign, an inversion-free image editing method for rectified flow models. Through terminal point regularization, the balance between semantic alignment and structural consistency can be improved. Extensive experiments illustrate the superior performance of the proposed method.

**Strengths:**

- The method is training-free and inversion-free, which can be applied on various off-the-shelf flow-based models.
- Compared to the baseline methods such as FlowEdit, the proposed method is more computationally efficient.
- The paper is well-written and easy to follow.

**Weaknesses:**

- The motivation of this paper is not clear.  Authors claim that 'these limitations of FlowEdit arise from non-smooth and unstable
editing trajectories' (Line 86). However, there is no experimental analysis in the paper to support this, making this claim unpromising.
- Authors claim that FlowEdit is sensitive to the hyperparameters. However, there are also many hyperparameters need to be tuned in the proposed method (including but not limited to the ω and ζ ). Could FlowAlign perform better than FlowEdit in terms of hyperparameter sensitivity?
- A comprehensive discussion of the related work is missing, there has emerged a number of image editing methods recently, such as RF-Edit [1], QK-Edit [2], FluxSpace [3], FireFlow [4]. Authors are highly suggested to discuss and compare the performance between the proposed method and these baselines.
- Authors say that 'FlowAlign achieves superior source preservation and competitive or improved editing quality compared to FlowEdit and also other existing approaches in image, video, and 3D editing.' However, they do not compare the performance of their method with any baselines of video editing. By the way, the prompt in Figure 13 ( A black swan is swimming...) does not match the video.

[1]. Taming Rectified Flow for Inversion and Editing

[2]. QK-Edit: Revisiting Attention-based Injection in MM-DiT for Image and Video Editing

[3]. FluxSpace: Disentangled Semantic Editing in Rectified Flow Transformers

[4]. FireFlow: Fast Inversion of Rectified Flow for Image Semantic Editing

**Questions:**

See weakness.

---

> ### Author Response · Authors · 2025-11-24
> **Response by Authors (1/2)**
>
> **W1. Missing motivation and experimental analysis that supports a smooth editing trajectory.**
>
> Contrary to the reviewer’s comment, we clearly explained that FlowEdit’s non-smooth editing causes loss of source consistency and requires heuristics such as early stopping, and that FlowAlign resolves these issues. The backward-editing experiment (Figure 5 in the original manuscript) further demonstrates FlowAlign’s ODE-like behavior (as pointed out by Reviewer 2ckP), as FlowEdit fails to reconstruct the source image.
>
> We additionally analyzed “smooth” editing using a toy 2D GMM setup (Figure 19 in the revised manuscript). FlowEdit often produces out-of-domain samples with curved, unstable transport trajectories (Figure 19-(c) and (c’)) while FlowAlign offers straight and smoother trajectories (Figure 19-(d) and (d’)). The optimal-control velocities in FlowAlign are clean and localized around the edit target, while FlowEdit’s velocities are noisy and dispersed. As a result, FlowEdit induces excessive and widespread deviations from the source, whereas FlowAlign preserves source proximity while applying the desired edit.
>
> These results provide clear empirical support for our motivation: FlowAlign achieves smoother, more stable editing trajectories through optimal control and source consistency regularization, as mentioned in the general comment G1.
>
>
> **W2. Missing hyperparameter sensitivity analysis**
>
> We would like to clarify that we have provided sensitivity analysis for hyper-parameters in our ablation study (Figure 6 in the original manuscript) where the same hyper-parameter $\zeta$ provides the balanced performance between semantic alignment and source consistency across the classifier-free guidance scale $\omega$. We also have emphasized that FlowAlign does not require early time stop as done in FlowEdit, which clearly reduces the sensitivity of hyper-parameters.
>
> Additionally we compare robustness of the proposed hyperparameters in FlowAlign and FlowEdit in Figure 22 in the revised manuscript. In FlowAlign (Figure 22-(a)), $\zeta=0.01$ shows robust performance on the trade-off between CLIP similarity and structural distance across various values of $\omega$. In contrast, the original **FlowEdit** formulation (Figure 22-(b)) lacks a clear Pareto optimal curve when evaluating the impact of its signal to noise ratio $N$. This suggests a non-robust relationship between $N$ and the editing outcomes. FlowEdit relies on a heuristically determined value (e.g., $N=0.34$ in $50$ timesteps) for early stopping, which is not systematically optimized for balancing these two objectives.

---

> ### Author Response · Authors · 2025-11-24
> **Response by Authors (2/2)**
>
> **W3. Missing comparison with related works.**
>
> Thank you for highlighting recent editing methods. We have added additional quantitative comparisons with suggested baselines in Table 1 and qualitative comparison in Figure 18 in the revised manuscript.
>
> We would like to clarify that RF-Edit [1], QK-Edit [2], FluxSpace [3], and FireFlow [4] all fundamentally rely on **source image inversion**, whereas **FlowAlign is fully inversion-free.** Our results show that FlowAlign achieves more faithful and controlled edits than the inversion-based baselines.
>
> Inversion-based methods inherently suffer from reconstruction errors, which lead to significant deviations from the source image. To mitigate this, RF-Edit, QK-Edit, and FireFlow inject attention-layer features (Q/K/V) extracted during the inversion process back into the generation process. FluxSpace leverages an editing-direction attention shift. However, all remain highly sensitive to inversion errors; even with higher-order ODE solvers, flow inversion is imperfect and computationally expensive. FlowEdit and FlowAlign bypasses inversion altogether, avoiding these errors without adding computational overhead, while FlowAlign further enhances source consistency with optimal control formulation and source consistency regularization.
>
> [1] Taming Rectified Flow for Inversion and Editing
>
> [2] QK-Edit: Revisiting Attention-based Injection in MM-DiT for Image and Video Editing
>
> [3] FluxSpace: Disentangled Semantic Editing in Rectified Flow Transformers
>
> [4] FireFlow: Fast Inversion of Rectified Flow for Image Semantic Editing
>
> **W4. Missing quantitative comparison for video and 3D GS editing.**
>
> We appreciate the reviewer for giving constructive feedback. We provide quantitative comparison for the video and 3D GS editings according to the reviewer’s request.
>
> For 3D GS Applications, we provide quantitative metrics in Table 2 in the general comment, comparing our method against score distillation methods (SDS, DDS, PDS) and IGS2GS. Across 15 scenes (6 distinct scenes with varying prompts), FlowAlign demonstrates superior performance not only in source consistency (measured by background LPIPS) but also in 3D consistency (measured by CLIP directional consistency) and human preference (measured by LAION aesthetic score).
>
> For Video Editing, quantitative comparison is presented in Table 3 in the general comment. We evaluated FlowAlign against FlowEdit and FlowDirector [5] (FlowEdit with attention masking mechanism), as FlowDirector is the only open-sourced training-free video editing algorithm up to our knowledge. We evaluated on the same baseline T2V model - Wan2.1T2V on 125 cases (25 videos from the DAVIS dataset with 5 different editing prompts). For fair comparison, we also include results for FlowAlign + masking. FlowAlign achieves superior source consistency compared to baselines while maintaining comparable text alignment results. Qualitative comparisons for video editing are provided in Figure 14 in the revised manuscript.
>
> Also, we fixed the typo in the caption.
>
>
> [5] FlowDirector: Training-Free Flow Steering for Precise Text-to-Video Editing

---

### Official Review · Reviewer_2ckP · 2025-11-04

**Soundness:** 3
**Presentation:** 3
**Contribution:** 3
**Rating:** 6
**Confidence:** 4

**Summary:**

This paper introduces FlowAlign, an inversion-free, flow-based framework for text-guided image editing built on pre-trained rectified flow models such as Stable Diffusion 3. It formulates editing as an optimal control problem and derives a modified velocity field that combines FlowEdit-style semantic drift with a source-consistency term expressed via Tweedie denoising of source and target latents. The resulting ODE is training-free, adds only a small computational overhead, and aims to produce more stable and deterministic editing trajectories. Experiments on PIEBench with SD3 show improved background preservation metrics and human preference over SDEdit, DDIB, RF-Inversion, and FlowEdit, while keeping CLIP and HPS scores competitive.

**Strengths:**

- The optimal-control formulation leads to a drift that explicitly combines semantic editing and source-consistency terms, offering a clear interpretation of how FlowAlign stabilizes trajectories.

- Using a shared SD3 backbone and comparable NFEs, FlowAlign consistently improves background PSNR/SSIM/LPIPS/DINO over SDEdit, DDIB, RF-Inversion, and FlowEdit while maintaining competitive CLIP and HPS scores.

- Reconstructing the original image by integrating backward yields much better fidelity with FlowAlign than with FlowEdit, reinforcing the claim of more stable, ODE-like behavior.

- Applying the trajectory regularizer as a guidance signal in these settings suggests that the idea could generalize beyond 2D image editing and inspire further work on structured generative representations.

**Weaknesses:**

- The optimal-control derivation relies on linear assumptions and a first-order approximation to the integrated drift, so the implemented ODE is not tightly characterized by theory and functions more as a guided heuristic.

- The paper positions FlowAlign mainly against multi-step ODE/SDE editors and does not clearly situate it relative to recent fast or one-step editing approaches that optimize for speed–quality trade-offs.

- Results on video and 3D Gaussian splatting are purely qualitative, without metrics for temporal or multi-view consistency, making it hard to assess robustness in these scenarios.

**Questions:**

How closely does the approximated drift in the implemented ODE track the exact optimal-control solution on simple toy problems, and did you observe regimes where the approximation qualitatively breaks down?

Are there characteristic failure modes (e.g., multi-object insertion, strong domain shifts) where FlowAlign underperforms FlowEdit or RF-Inversion, and could you provide concrete examples or diagnostics?

To what extent can the core idea be ported to non-rectified-flow backbones e.g. standard diffusion models, and what modifications would be required?

---

> ### Author Response · Authors · 2025-11-24
> **Response by Authors (1/3)**
>
> **W1. Error from the first-order approximation in the optimal control derivation.**
>
> Thank you for the constructive comment. We agree that the first order approximation $v_t = -t v_t(p_t) + t v_t(q_t)$ introduces error, particularly for larger $t$ near 1. Although this is not an exact integration of the ODE, it is a standard first-order Euler (rectangle-rule) approximation of the integration. Under typical Lipschitz continuity assumptions, the local truncation error scales as $O(t^2)$, which is the expected behavior of the first-order ODE discretizations and is widely adopted in diffusion sampling when using Euler solver.
>
> Nevertheless, this approximation can lead to occasional failure cases at the initial step near t=1, as shown in our response for the Q1. However, such failure cases occur less frequently than with FlowEdit. Empirically, we found that this first-order approximation is sufficiently accurate for our purposes and provides favorable computational efficiency.
>
> **W2. Missing comparison with one-step editing approaches.**
>
> We would like to assure the reviewer that the scope of this paper is “leveraging noise-to-image flow models for image-to-image editing without any additional training”, rather than designing accelerated editing methods. Although faster or even one-step editing approaches exist, they typically require task-specific training or fine-tuning using conditional flow matching, where the source image is treated as an additional conditioning signal. Therefore, comparison with such one-step editing methods fall outside the scope of our work.
>
> **W3. Purely qualitative results on video and 3D GS editing are provided.**
>
> Thank you for pointing out missing quantitative results in our additional applications with video and 3D GS. Per reviewer’s request, we measure the quantitative metrics for each application.
>
> For 3D GS Applications, we provide quantitative metrics in Table 2, comparing our method against score distillation methods (SDS, DDS, PDS) and IGS2GS. Across 15 scenes (6 distinct scenes with varying prompts), FlowAlign demonstrates superior performance not only in source consistency (measured by background LPIPS) but also in 3D consistency (measured by CLIP directional consistency) and human preference (measured by LAION aesthetic score).
>
> For Video Editing, quantitative comparison is presented in Table 3. We evaluated FlowAlign against FlowEdit and FlowDirector [1] (FlowEdit with attention masking mechanism) on 125 cases (25 videos from the DAVIS dataset with 5 different editing prompts). We also include results for FlowAlign + masking. FlowAlign achieves superior source consistency compared to baselines while maintaining comparable text alignment results. Qualitative comparisons for video editing are provided in Figure 14.
>
> [1] FlowDirector: Training-Free Flow Steering for Precise Text-to-Video Editing

---

> ### Author Response · Authors · 2025-11-24
> **Response by Authors (2/3)**
>
> **Q1. Approximation error analysis between implemented ODE and exact optimal-control solution with toy example**
>
> To provide empirical evidence for the efficacy of the proposed regularization term, we validate our algorithm using a controlled 2D toy experiment. As illustrated in Figure 19-(a) in the revised manuscript, the toy experiment utilizes Gaussian Mixture Models distributed uniformly in 2D space to represent the source data (green), target data (blue), and noise (red) distributions. Since direct source and target prompt conditioning is infeasible in this setting, separate Rectified Flow models were trained for the noise-to-source and noise-to-target data mappings.
>
> We compare the transport performance across three methodologies:
> - 1.	Inversion and Sampling: Figure 19-(b) and 19-(b') demonstrate that the sequential application of source-to-noise inversion followed by noise-to-target sampling successfully transports data from the source domain to the target domain along clear trajectories.
> - 2.	FlowEdit: As FlowEdit samples random noise at every time step, this random noise selection frequently leads to out-of-domain outputs (Figure 19-(c)) and results in non-straight transport trajectories (Figure 19-(c')).
> - 3.	FlowAlign: Figures 19-(d) illustrates the effect of incorporating our regularization term, which is derived from the first-order approximation of the optimal control solution. This regularization significantly reduces the prevalence of out-of-domain samples (Figure 19-(d)) compared to FlowEdit. Crucially, FlowAlign yields straighter transport paths (Figure 19-(d')) and effectively ensures that source data is transported to close target data, simplifying the complex trajectory in Figure 19-(b').
>
> The reliance on the first-order velocity approximation and the multi-step nature of the flow-based model introduces some failure cases. As depicted in Figure 19-(d''), when the predicted velocity field points in an incorrect direction during the initial timesteps (low signal-to-noise ratio; purple), the generated sample trajectory can diverge (ending in yellow), leading to out-of-domain samples. However, we like to assure the reviewer that this occurrence is observed less frequently in FlowAlign (Figure 19-(d)) compared to FlowEdit (Figure 19-(c)).

---

> ### Author Response · Authors · 2025-11-24
> **Response by Authors (3/3)**
>
> **Q2. Failure cases**
>
> In response to the request for failure cases, especially compared to FlowEdit and RF-Inversion, Table 4 presents the quantitative comparison of CLIP scores across all nine PIEBench categories for the proposed FlowAlign method and selected baselines. While FlowAlign achieves competitive performance across most categories, the method exhibits its lowest CLIP score in the "3-delete object" category. Detailed qualitative examples illustrating these limitations are provided in Figure 21 in the revised manuscript.
> The challenge of precise object deletion often results in FlowAlign generating an unwanted replacement in the original object's location. This issue is not unique to our approach; both FlowEdit and SDEdit exhibit similar failure modes. Furthermore, FlowEdit frequently compromises overall image fidelity by introducing unintended transformations to regions outside the specified editing area. In contrast, while RF-Inversion and DDIB more closely adhere to the textual editing instructions for deletion, they fail to maintain satisfactory source image consistency, leading to significant content deviation from the original image.
>
> *Table 4. CLIP score on all PIEBench Categories*
>
> | method       | 0-random | 1-change object | 2-add object | **3-delete object** | 4-change attribute content | 5-change attribute pose | 6-change attribute color | 7-change attribute material | 8-change background | 9-change style |
> | ------------ | -------- | --------------- | ------------ | ------------------- | -------------------------- | ----------------------- | ------------------------ | --------------------------- | ------------------- | -------------- |
> | DDIB         | 26.72    | 23.95           | 25.46        | 22.50               | 23.44                      | 23.66                   | 24.89                    | 24.13                       | 24.03               | 25.30          |
> | FlowEdit     | 25.48    | 26.44           | 27.18        | 24.82               | 26.28                      | 26.21                   | 26.52                    | 26.39                       | 25.98               | 28.14          |
> | RF-Inversion | 24.77    | 27.33           | 28.25        | 26.50               | 26.63                      | 26.96                   | 28.70                    | 27.81                       | 27.16               | 29.59          |
> | SDEdit       | 26.08    | 24.081          | 25.87        | 23.46               | 24.32                      | 25.241                  | 24.66                    | 24.88                       | 24.45               | 26.83          |
> | Ours         | 27.41    | 25.30           | 26.42        | 23.69               | 25.20                      | 25.75                   | 26.19                    | 25.98                       | 25.47               | 27.22          |
>
> **Q3. Extend to non-rectified flow backbones.**
>
> Since flow models are a general framework that includes denoising diffusion models as a special case, FlowAlign can be applied to any non-rectified flow backbones by using an appropriate change of variables to convert $\epsilon$-prediction into velocity prediction.

---

### Author Response · Authors · 2025-11-24
**General comments by Authors**

We sincerely appreciate the reviewers’ time and effort in providing valuable comments on our submission. We are particularly encouraged by the recognition that our work **offers a clear interpretation of smoother editing trajectory through optimal control theory, achieves higher fidelity compared to FlowEdit** (2ckP, JBzF), and **provides computational efficiency** (GZwn, JBzF) **while clearly outperforming the baselines** (2ckP, TFcK). Below, we address each of the reviewers’ concerns point by point. Before that, we would like to highlight a general response to the major concerns commonly raised across the reviews.

**G1. Motivation and intuition of FlowAlign**

A main contribution of FlowAlign is introducing optimal control-based interpretation of inversion-free image editing and source consistency regularization that allows a smoother editing trajectory. This improves editing outcomes in two key aspects.

First, it prevents excessive deviation from the original generative path - an issue introduced by DDPM-inversion-style noise injection [1]. Although this noise enables flexible image editing, it often causes overly saturated results unless parameters such as classifier-free guidance scale or early stopping are carefully tuned, as in FlowEdit. FlowAlign mitigates these issues naturally, without relying on such heuristics.

Second, a smoother trajectory strengthens source consistency. While baseline methods including FlowEdit adhere well to editing instructions, it often loses key source attributes (e.g. identity or background) due to the absence of an explicit consistency mechanism. FlowAlign introduces a simple, theoretically grounded regularization term that preserves source information while maintaining strong editing capability.

[1] An Edit Friendly DDPM Noise Space: Inversion and Manipulations, CVPR 2024

**G2. Missing comparison with related works.**

We thank all reviewers for their constructive comments on missing related works. For the 2D image editing, we provide additional comparison results with relevant editing approaches that leverage inversion in Table 1, while the proposed method is inversion-free editing approach. Also, for 3DGS and video editing applications, we provide quantitative comparison with suggested baseline methods in Table 2&3. All of these results support that FlowAlign shows superior source consistency with comparable text alignment, and applicability to various domains (3D, video) or pre-trained models (FLUX).

*Table 1. Quantitative Comparison on additional image editing baselines on FLUX*

|                | structure  | (background) psnr | (background) lpips | (background) mse | (background) ssim | CLIP (target) |
| -------------- | ---------- | ----------------- | ------------------ | ---------------- | ----------------- | ------------- |
| fireflow       | 0.0456     | 20.37             | 0.1824             | 0.0136           | 0.7584            | 26.69         |
| fluxspace       | 0.1647     | 11.69             | 0.3914             | 0.0971           | 0.5747            | 24.41         |
| rf-solver-edit | 0.0451     | 20.34             | 0.1824             | 0.0136           | 0.7589            | 26.75         |
| FlowAlign      | **0.0414** | **20.75**         | **0.1314**         | **0.0122**       | **0.8123**        | **26.97**     |


*Table 2. Quantitative Comparison on 3DGS editing*

|           | (background) LPIPS | CLIP Directional Similarity | CLIP     | Aesthetic Score |
| --------- | ------------------ | --------------------------- | -------- | --------------- |
| SDS       | 0.33               | 0.10                        | **0.27** | 4.94            |
| DDS       | 0.33               | 0.18                        | 0.27     | 4.76            |
| PDS       | 0.23               | 0.12                        | 0.24     | 4.12            |
| IGS2GS    | 0.10               | 0.07                        | 0.23     | 5.48            |
| FlowAlign | **0.06**           | **0.18**                    | 0.26     | **5.52**        |

*Table 3. Quantitative Comparison on video editing*

|                        | (background) psnr | (background) lpips | (background) mse | (background) ssim | overall_consistency | CLIP (target) |
| ---------------------- | ----------------- | ------------------ | ---------------- | ----------------- | ------------------- | ------------- |
| FlowEdit               | 26.73             | 0.0719             | 0.0032           | 0.81670           | 0.2164              | 24.18         |
| FlowAlign              | **27.19**         | **0.0662**         | **0.0028**       | 0.8286            | 0.2209              | 24.22         |
| FlowDirector (FE+mask) | 25.39             | 0.0765             | 0.0043           | 0.8197            | **0.2386**          | **25.41**     |
| FlowAlign + mask       | 26.46             | 0.0702             | 0.0037           | **0.8340**        | 0.2276              | 24.85         |

---

### Meta-Review · Area_Chair_o9oU · 2025-12-13

**Summary:**

This paper proposes FlowAlign, an inversion free, training free flow based image editing method that regularizes the editing trajectory via an optimal control view with a terminal source consistency term. The initial reviews were split: two reviewers (2ckP, TFck) were marginally positive (6), while two (GZwn, JBzF) were negative (2), largely due to questions about motivation, missing experiments, and missing comparisons.

After reading the full rebuttal and discussion, I think the authors did a solid job addressing the main actionable concerns. They added a toy 2D experiment to support the “trajectory smoothness” motivation, expanded comparisons to recent inversion based rectified flow editors, provided quantitative evaluations for video and 3D Gaussian splatting applications, and added hyperparameter robustness analysis clarifying that FlowAlign does not rely on FlowEdit style early stopping. Importantly, one of the strongest negative points raised, the request to compare with FlowCycle, is based on a post deadline arXiv work and should not be treated as a valid basis for rejection under standard review guidelines.

From the AC perspective, I view this as a clear and practically useful improvement over FlowEdit that strengthens source consistency while keeping the method simple and efficient. While the derivation involves approximations, the empirical evidence is now broad and consistent across multiple settings, and the contribution is well aligned with the current direction of training free editing for flow models. I recommend acceptance.

**Reviewer Concerns:**

Concerns addressed by the rebuttal:

1. Motivation and evidence for smoother trajectories improving editing: Added a controlled toy 2D experiment plus additional discussion and diagnostics that directly support the claim.

2. Missing related works and baseline comparisons: Added quantitative comparisons against several recent rectified flow inversion based editors (e.g., FireFlow, FluxSpace, RF Solver Edit), clarifying scope and positioning as inversion free.

3. Lack of quantitative evaluation for video and 3DGS editing: Added metrics and comparisons in both settings, moving these sections beyond purely qualitative evidence.

4. Hyperparameter sensitivity and reliance on heuristics: Added robustness analysis and clarified FlowAlign does not use early timestep skipping, reducing heuristic dependence compared with FlowEdit.

5. Failure modes: Added category level analysis (notably object deletion) and qualitative examples, clarifying expected limitations.

Concerns that remain but are not blocking:

1. Approximation in the optimal control derivation: The theory is not a tight characterization of the implemented solver, but the paper is transparent about the approximation and now backs the motivation with targeted experiments.

2. Absolute editing strength tradeoff: Some edits like deletion remain challenging, but this is common across baselines and the paper frames this fairly.

**Reviewer Scores:**

Reviewer 2ckP: likely remains 6, possibly slightly higher, since the rebuttal directly addressed the theory verification request and added quantitative results for additional domains.

Reviewer TFck: likely remains 6, as the rebuttal clarified the intuition, figure notation, and early skipping question.

Reviewer GZwn: could increase from 2 to around 4, since most concrete weaknesses (motivation evidence, missing baselines, video metrics, robustness) were directly addressed.

Reviewer JBzF: likely increase to 6, as the central negative argument relies heavily on a post deadline comparison request.

Overall, despite one reviewer remaining unconvinced, the rebuttal substantially strengthened the empirical case and resolved several key factual and experimental gaps. Considering the practicality of the contribution and the clarified positioning, my final recommendation is accept.

---

### Decision · Program_Chairs · 2026-01-26

Accept (Poster)